# iFlow: Numerically Invertible Flows for Efficient Lossless Compression via a Uniform Coder

**Shifeng Zhang, Ning Kang, Tom Ryder, Zhenguo Li**
Huawei Noah's Ark Lab
{zhangshifeng4, kang.ning2, tom.ryder1, li.zhenguo}.huawei.com

## Abstract

It was estimated that the world produced $59ZB$ ($5.9 \times 10^{13}GB$) of data in 2020, resulting in the enormous costs of both data storage and transmission. Fortunately, recent advances in deep generative models have spearheaded a new class of so-called "neural compression" algorithms, which significantly outperform traditional codecs in terms of compression ratio. Unfortunately, the application of neural compression garners little commercial interest due to its limited bandwidth; therefore, developing highly efficient frameworks is of critical practical importance. In this paper, we discuss lossless compression using normalizing flows which have demonstrated a great capacity for achieving high compression ratios. As such, we introduce iFlow, a new method for achieving efficient lossless compression. We first propose Modular Scale Transform (MST) and a novel family of numerically invertible flow transformations based on MST. Then we introduce the Uniform Base Conversion System (UBCS), a fast uniform-distribution codec incorporated into iFlow, enabling efficient compression. iFlow achieves state-of-the-art compression ratios and is $5\times$ quicker than other high-performance schemes. Furthermore, the techniques presented in this paper can be used to accelerate coding time for a broad class of flow-based algorithms.

## 1 Introduction

The volume of data, measured in terms of IP traffic, is currently witnessing an exponential year-on-year growth [13]. Consequently, the cost of transmitting and storing data is rapidly becoming prohibitive for service providers, such as cloud and streaming platforms. These challenges increasingly necessitate the need for the development of high-performance lossless compression codecs.

One promising solution to this problem has been the development of a new class of so-called "neural compression" algorithms [30, 38, 18, 4, 17, 37, 26, 31, 6, 41]. These methods typically posit a deep probabilistic model of the data distribution, which, in combination with entropy coders, can be used to compress data with the minimal codelength bounded by the negative log-likelihood [29]. However, despite reliably improved compression performance compared to traditional codecs [14, 33, 9, 34], meaningful commercial applications have been limited by impractically slow coding speed.

In this paper, we focus on developing approaches with deep probabilistic models based on *normalizing flows* [11, 25, 32, 27, 7, 28, 23]. A normalizing flow admits a learnable bijective mapping between input data and a latent variable representation. In this paradigm, inputs can be compressed by first transforming data to latent variables, with the resulting output encoded by a prior distribution. Compared to other classes of generative models, normalizing flows typically perform best in both tasks of probability density estimation (as compared with variational autoencoders [24, 15, 8]) and inference speed (as compared with autoregressive factorizations [35, 39, 22]). This suggests that compression with flows can jointly achieve high compression ratios along with fast coding times.

35th Conference on Neural Information Processing Systems (NeurIPS 2021).

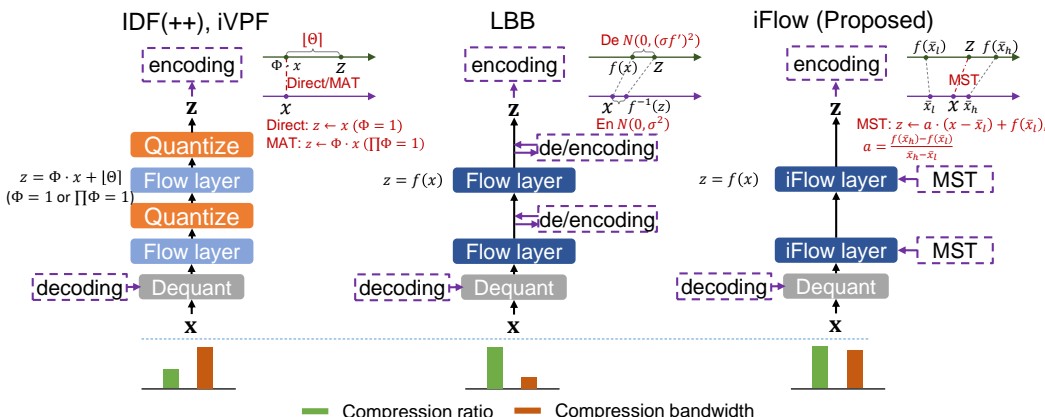

Figure 1: Illustration of IDF(++), iVPF, LBB and the proposed iFlow method. Flow layers with darker color denote higher expressive power. The top-right of each model illustration shows the key procedure of exact bijections between $\bar{x}$ and $\bar{z}$. The compression ratio and bandwidth are listed below.

Unfortunately, lossless compression requires discrete data for entropy coding, and the continuous bijections of normalizing flows would not guarantee discrete latent variables. As such, transformations would require discretization, resulting in a loss of information [41, 3]. To resolve this issue, Integer Discrete Flows (IDF) [18, 4] (Fig. 1, left) proposed an invertible mapping between discrete data *and* latent variables. Similarly, iVPF [41] achieves a discrete-space bijection using *volume-preserving* flows. However, the above models must introduce constraints on flow layers to ensure a discrete-space bijection, which limits the expressivity of the transformation. Local Bits-Back Coding (LBB) [17] (Fig. 1, middle) was the first approach to succeed in lossless coding with the flexible family of continuous flows. It resolves the information loss by coding numerical errors with the rANS coder [12]. However, the coder is extraordinarily slow, making the method impractical.

In this paper, we introduce iFlow: a numerically invertible flow-based neural compression codec that bridges the gap between high-fidelity density estimators and practical coding speed. To achieve this we introduce two novelties: Modular Scale Transform (MST) and Uniform Base Conversion Systems (UBCS). MST presents a flexible family of bijections in discrete space, deriving an invertible class of flows suitable for lossless compression. UBCS, built on the uniform distribution, permits a highly efficient entropy coding compatible with MST. The main ideas are illustrated in Fig. 1(right). We test our approach across a range of image datasets against both traditional and neural compression techniques. Experimentally we demonstrate that our method achieves state-of-the-art compression ratios, with coding time $5\times$ quicker than that of the next-best scheme, LBB.

## 2 Numerically Invertible Flows and Lossless Compression

Let $f : \mathcal{X} \rightarrow \mathcal{Z}$ be our normalizing flow, which we build as a composition of layers such that $f = f_L \circ ... \circ f_2 \circ f_1$. Defining $d$-dimensional $\mathbf{y}_0 = \mathbf{x} \in \mathcal{X}$ and $\mathbf{y}_L = \mathbf{z} \in \mathcal{Z}$, the latents can be computed as $\mathbf{y}_l = f_l(\mathbf{y}_{l-1})$, with $p_X(\mathbf{x})$ calculated according to

$$\log p_X(\mathbf{x}) = \log p_Z(\mathbf{z}) + \sum_{l=1}^{L} \log |\det J_{f_l}(\mathbf{y}_{l-1})|, \tag{1}$$

where $J_{f_l}(\mathbf{x})$ is the Jocabian matrix of the transformation $f_l$ at $\mathbf{y}_{l-1}$. In this paper, we use $\log$ to denote *base 2 logarithms*. Training the flow should minimize the negative log-likelihood $-\log p_X(\mathbf{x})$. The inverse flow is defined as $f^{-1} = f_1^{-1} \circ ... \circ f_L^{-1}$, such that $f^{-1}(\mathbf{z}) = \mathbf{x}$. There are many types of invertible bijections, $f_l$. Popular choices include element-wise, autoregressive (e.g. coupling layers) [41] and $1 \times 1$ convolution transformations [25].

To perform lossless compression with normalizing flows, the input data should be transformed to latent variables, which are then encoded with a prior distribution. As data and encoded bits should be in binary format, $\mathbf{x}$ and $\mathbf{z}$ must be discrete, with a bijection between discrete inputs and outputs

established. However, as discussed, this operation is usually intractable with popular classes of transformations used in continuous flows due to the numerical errors induced by discretization.

In this section, we introduce a novel algorithm derived from continuous flows that allows us to achieve an exact bijective mapping between discrete $\mathbf{x}$ *and* discrete $\mathbf{z}$. We present the general idea of a flow layer's numerical invertibility in discrete space in Sec. 2.1. We then introduce our flow layers in Sec. 2.2 and 2.3, with their application to compression discussed in Sec. 2.4 and 2.5.

## 2.1 Invertibility of Flows in Discrete Space

To discretize the data, we follow [41] and adopt *k-precision quantization* to assign floating points to discretization bins such that

$$\bar{x} = \frac{\lfloor 2^k \cdot x \rfloor}{2^k}, \tag{2}$$

where $\lfloor \cdot \rfloor$ is the floor function. The associated quantization error is bounded by $|\bar{x} - x| < 2^{-k}$. Any $d$-dimensional $\mathbf{x}$ can be quantized into bins with volume $\delta = 2^{-kd}$. The probability mass of $\bar{\mathbf{x}}$ can then be approximated by $P(\bar{\mathbf{x}}) = p(\bar{\mathbf{x}})\delta$, such that the theoretical codelength is $-\log(p(\bar{\mathbf{x}})\delta)$.

Denote our proposed numerically invertible flow (iFlow) according to $\bar{f} = \bar{f}_L \circ ... \circ \bar{f}_1$. The input and output should be subject to $k$-precision quantization, where each layer should be appropriately invertible such that $\bar{\mathbf{y}}_{l-1} \equiv \bar{f}_l^{-1}(\bar{f}_l(\bar{\mathbf{y}}_{l-1}))$. Denoting $\bar{\mathbf{y}}_0 = \mathbf{x}$ and $\bar{\mathbf{z}} = \bar{\mathbf{y}}_L$, it is further expected that $\bar{\mathbf{z}} = \bar{f}(\mathbf{x})$ and $\mathbf{x} \equiv \bar{f}^{-1}(\bar{\mathbf{z}})$. We will show in the following subsections that $\bar{f}$ can be derived from *any* continuous flow $f$ in such a way that the error between $\bar{f}(\mathbf{x})$ and $f(\mathbf{x})$ is negligible.

## 2.2 Numerically Invertible Element-Wise Flows

We begin by describing our approach for element-wise flows $\mathbf{z} = f(\mathbf{x})$, where $f$ is a monotonic element-wise transformation. For notation simplicity, we assume that the flow contains just one layer. In most cases, $f$ does not present a unique inverse in discrete space, such that $f^{-1}(\bar{\mathbf{z}}) \neq \mathbf{x}$. In what follows, we introduce an approach to derive an invertible, discrete-space operation from any element-wise flow transformation. For simplicity, we denote the inputs and output pairs of a continuous flow as $x$ and $z = f(x)$; and we denote the $k$-precision quantized input-output pairs of iFlow as $\bar{x}$ and $\bar{z} = \bar{f}(\bar{x})$.

### 2.2.1 Scale Flow

---

**Algorithm 1** Modular Scale Transform (MST): Numerically Invertible Scale Flow $f(x) = R/S \cdot x$.

---

**Forward MST:** $\bar{z} = \bar{f}(\bar{x})$.      **Inverse MST:** $\bar{x} = \bar{f}^{-1}(\bar{z})$.

1: $\hat{x} \leftarrow 2^k \cdot \bar{x}$;                                      1: $\hat{z} \leftarrow 2^k \cdot \bar{z}$;
2: Decode $r_d$ using $U(0, R)$; $\hat{y} \leftarrow R \cdot \hat{x} + r_d$;     2: Decode $r_e$ using $U(0, S)$; $\hat{y} \leftarrow S \cdot \hat{z} + r_e$;
3: $\hat{z} \leftarrow \lfloor \hat{y}/S \rfloor, r_e \leftarrow \hat{y} \mod S$;             3: $\hat{x} \leftarrow \lfloor \hat{y}/R \rfloor, r_d \leftarrow \hat{y} \mod R$;
4: Encode $r_e$ using $U(0, S)$;                       4: Encode $r_d$ using $U(0, R)$;
5: **return** $\bar{z} \leftarrow \hat{z}/2^k$.                          5: **return** $\bar{x} \leftarrow \hat{x}/2^k$.

---

We begin with the scale transformation defined as $z = f(x) = a \cdot x, (a > 0)$, from which we are able to build more complex flow layers. The input can be converted to an integer with $\hat{x} = 2^k \cdot \bar{x}$, and the latent variable $\bar{z}$ can be recovered from integer $\hat{z}$ by $\bar{z} = \hat{z}/2^k$. Inspired by MAT in iVPF [41], we first approximate $a$ with a fractional such that $a \approx \frac{R}{S}$, where $R, S \in \mathbb{N}$. Denote $\hat{y} = R \cdot \hat{x} + r_d$ where $r_d$ is sampled uniformly from $\{0, 1, ..., R-1\}$, we can obtain $\hat{z}$ with $\hat{z} = \lfloor \hat{y}/S \rfloor$, and the remainder $r_e = \hat{y} \mod R$ can be encoded to eliminate the error. It is clear that $(\hat{x}, r_d) \leftrightarrow \hat{y} \leftrightarrow (\hat{z}, r_e)$ are bijections.

Let $U(0, A)$ be the uniform distribution such that $P(s) = \frac{1}{A}$, $s \in \{0, 1, ..., A-1\}$. The numerically invertible scale flow $\bar{f}$ is displayed in Alg. 1. As the modular operation is essential to Alg. 1, we name it the *Modular Scale Transform* (MST). Setting $S$ and $R = \text{round}(S \cdot a)$ to be large, we observe that the following propositions hold.

**Proposition 1.** *[41]* $|\bar{z} - f(\bar{x})| < O(S^{-1}, 2^{-k})$.

**Proposition 2.** *The codelength of MST is* $L_f(\bar{x}, \bar{z}) = \log S - \log R \approx -\log |a| = -\log |f'(\bar{x})|$.

Proposition 1 establishes that the error is small if $S$ and $k$ are large. Proposition 2 demonstrates that the codelength is almost exactly the log-Jocabian of $f$. The above properties and the correctness of the algorithm are discussed in the Appendix. Further note that, with the exception of the usual encoding and decoding processes, MST's operations can be parallelized, resulting in minimal additional overhead. In contrast, MAT's operation in iVPF [41] only deals with volume-preserving affine transform and is performed sequentially along dimensions, limiting its usage and efficiency.

### 2.2.2 General Element-wise Flow

---

**Algorithm 2** Numerically Invertible Element-wise Flows

---

**Forward:** $\bar{z} = \bar{f}(\bar{x})$.

1: Get $\bar{x}_l, \bar{x}_h, \bar{z}_l, \bar{z}_h$ given $\bar{x}$, so that $\bar{x} \in [\bar{x}_l, \bar{x}_h)$;
2: Set large $S$; get $R$ with Eq. (4); $\bar{x}' \leftarrow \bar{x} - \bar{x}_l$;
3: Get $\bar{z}'$ with forward MST in Alg. 1, given input $\bar{x}'$ and coefficients $R, S$;
4: **return** $\bar{z} = \bar{z}' + \bar{z}_l$.        $\triangleright \bar{z} \in [\bar{z}_l, \bar{z}_h)$.

**Inverse:** $\bar{x} = \bar{f}^{-1}(\bar{z})$.

1: Get $\bar{x}_l, \bar{x}_h, \bar{z}_l, \bar{z}_h$ given $\bar{z}$, so that $\bar{z} \in [\bar{z}_l, \bar{z}_h)$;
2: Set large $S$; get $R$ with Eq. (4); $\bar{z}' \leftarrow \bar{z} - \bar{z}_l$;
3: Get $\bar{x}'$ with inverse MST in Alg. 1, given input $\bar{z}'$ and coefficients $R, S$;
4: **return** $\bar{x} = \bar{x}' + \bar{x}_l$.        $\triangleright \bar{x} \in [\bar{x}_l, \bar{x}_h)$.

---

For simplicity, we assume the non-linear $f$ is monotonically increasing. For a monotonically decreasing $f$, we simply invert the sign: $-f$.

MST in Alg. 1 works with *linear* transformations, and is incompatible with *non-linear* functions. Motivated by linear interpolation, we can approximate the non-linear flow with a piecewise linear function. In general, consider input $\bar{x}$ to be within an interval $\bar{x} \in [\bar{x}_l, \bar{x}_h)^1$ and $\bar{z}_l = \overline{f(\bar{x}_l)}, \bar{z}_h = \overline{f(\bar{x}_h)}$. The linear interpolation of $f$ is then

$$f_{inp}(x) = \frac{\bar{z}_h - \bar{z}_l}{\bar{x}_h - \bar{x}_l}(\bar{x} - \bar{x}_l) + \bar{z}_l. \tag{3}$$

It follows that $\bar{z}$ can be derived from MST with input $\bar{x} - \bar{x}_l$ followed by adding $\bar{z}_l$. Furthermore, $\bar{z}$ and $\bar{x}$ can be recovered with the corresponding inverse transformation. To preserve monotonicity, $\bar{z}$ must be within interval $\bar{z} \in [\bar{z}_l, \bar{z}_h)$. Denoting the invertible linear flow derived from $f_{inp}$ as $\bar{f}_{inp}$, it must hold that $\bar{f}_{inp}(\bar{x}_l) \geq \bar{z}_l, \bar{f}_{inp}(\bar{x}_h - 2^{-k}) \leq \bar{z}_h - 2^{-k}$. As we use MST in Alg. 1, we observe that when $\frac{\bar{z}_h - \bar{z}_l}{\bar{x}_h - \bar{x}_l}$ is approximated with $R/S$, the minimum possible value of $\bar{f}_{inp}(\bar{x}_l)$ is $\bar{z}_l$ (when $r_d = 0$ in Line 2) and the maximum possible value of $\bar{f}_{inp}(\bar{x}_h - 2^{-k})$ is $\lfloor (R \cdot 2^k (\bar{x}_h - \bar{x}_l - 2^{-k}) + R - 1)/(S \cdot 2^k) \rfloor + \bar{z}_l$ (when $r_d = R - 1$ in Line 2). Given large $S$, the largest possible value of $R$ should be

$$R = \lfloor \frac{(2^k \cdot (\bar{z}_h - \bar{z}_l) - 1) \cdot S + 1}{2^k \cdot (\bar{x}_h - \bar{x}_l)} \rfloor. \tag{4}$$

Another difficulty is in determining the correct interpolation interval $[\bar{x}_l, \bar{x}_h)$. One simple solution is to split the input domain into uniform intervals with length $2^{-h}$ such that $\bar{x}_l = \lfloor (2^h \cdot \bar{x})/2^h \rfloor, \bar{x}_h = \bar{x}_l + 2^{-h}$. However, for the inverse computation given $\bar{z}$, it is not easy to recover the interpolation interval as $f^{-1}(\bar{z})$ may not be within $[\bar{x}_l, \bar{x}_h)$. Instead, as $\bar{z} \in [\bar{z}_h, \bar{z}_l)$, the interval $[\bar{x}_l, \bar{x}_h)$ can be obtained via a binary search in which $\bar{x}_l$ is obtained with $\bar{z} \in [\overline{f(\bar{x}_l)}, \overline{f(\bar{x}_h)})$. Another approach is to split the co-domain into uniform intervals such that $\bar{z}_l = \lfloor (2^h \cdot \bar{z})/2^h \rfloor, \bar{z}_h = \bar{z}_l + 2^{-h}$, with $\bar{x}_l = \overline{f^{-1}(\bar{z}_l)}, \bar{x}_h = \overline{f^{-1}(\bar{z}_h)}$. In this case, determining $[\bar{z}_l, \bar{z}_h)$ during the inverse computation is simple. While for the forward pass, $[\bar{z}_l, \bar{z}_h)$ should be determined with a binary search such that $\bar{x} \in [\overline{f^{-1}(\bar{z}_l)}, \overline{f^{-1}(\bar{z}_h)})$. In practice, we have simpler tricks to determine the correct interval $[\bar{x}_l, \bar{x}_h), [\bar{z}_l, \bar{x}_h)$ for both the forward and inverse computation, which can be found in the Appendix.

The general idea of the non-linear flow adaptations are summarized in Alg. 2. Note that Alg. 2 can be used in any element-wise flow including linear flow. In Alg. 2, Proposition 2 holds such that $L_f(\bar{x}, \bar{z}) = -\log(R/S) \approx -\log \frac{\bar{z}_h - \bar{z}_l}{\bar{x}_h - \bar{x}_l} \approx -\log |f'(\bar{x})|$. It is possible to arrive at a similar conclusion in Proposition 1 where the corresponding error is $|\bar{z} - f(\bar{x})| < O(S^{-1}, 2^{-k}, 2^{-2h})$.

---

[1] All intervals must partition the domain of $f$.

## 2.3 Practical Numerically Invertible Flow Layers

Whilst one can use many types of complex transformations to build richly expressive flow-based models, to ensure the existence and uniqueness of the inverse computation, these layers are generally constructed with element-wise transformations. In this subsection, we demonstrate the invertibility of discretized analogs to some of the most widely used flow layers using the operators as described in Sec. 2.2. Such flows include autoregressive flows [20] (including coupling flows [10, 11, 16]) and $1 \times 1$ convolutional flows [25].

### 2.3.1 Autoregressive and Coupling Flows

---

**Algorithm 3** Numerically Invertible Autoregressive Flow

---

| **Forward:** $\bar{\mathbf{z}} = \bar{f}(\bar{\mathbf{x}})$. | **Inverse:** $\bar{\mathbf{x}} = \bar{f}^{-1}(\bar{\mathbf{z}})$. |
|---|---|
| 1: **for** $i = m, ..., 1$ **do** | 1: **for** $i = 1, ..., m$ **do** |
| 2:    $\bar{\mathbf{z}}_i \leftarrow \bar{f}_i(\bar{\mathbf{x}}_i, \bar{\mathbf{x}}_{<i})$ with Alg. 2 (forward); | 2:    $\bar{\mathbf{x}}_i \leftarrow \bar{f}_i^{-1}(\bar{\mathbf{z}}_i, \bar{\mathbf{x}}_{<i})$ with Alg. 2 (inverse); |
| 3: **end for** | 3: **end for** |
| 4: **return** $\bar{\mathbf{z}} = [\bar{\mathbf{z}}_1, ..., \bar{\mathbf{z}}_m]^\top$. | 4: **return** $\bar{\mathbf{z}} = [\bar{\mathbf{z}}_1, ..., \bar{\mathbf{z}}_m]^\top$. |

---

Supposing that the inputs and outputs, $\mathbf{x}$ and $\mathbf{z}$, are split into $m$ parts $\mathbf{x} = [\mathbf{x}_1, ..., \mathbf{x}_m]^\top, \mathbf{z} = [\mathbf{z}_1, ..., \mathbf{z}_m]^\top$, the autoregressive flow $\mathbf{z} = f(\mathbf{x})$ can be represented as $\mathbf{z}_i = f_i(\mathbf{x}_i; \mathbf{x}_{<i})$, where $f_i(\cdot; \mathbf{x}_{<i})$ is the element-wise flow (discussed in Sec. 2.2), conditioned on $\mathbf{x}_{<i}$. Now let $\bar{f}_i(\cdot; \mathbf{x}_{<i})$ denote the invertible element-wise flow transformation as discussed in Alg. 1-2. Alg. 3 then illustrates the details of an invertible autoregressive flow $\bar{f}$.

Propositions 1 and 2 hold in our discretized autoregressive transformation. In fact, the log-determinant of Jacobian is given by $\log|\det J_f(\mathbf{x})| = \sum_{i=1}^m \log|\det J_{f_i}(\mathbf{x}_i; \mathbf{x}_{<i})|$, and the expected codelength is simply $L_f(\bar{\mathbf{x}}, \bar{\mathbf{z}}) = \sum_{i=1}^m L_f(\bar{\mathbf{x}}_i, \bar{\mathbf{z}}_i) \approx -\sum_{i=1}^m \log|\det J_{f_i}(\bar{\mathbf{x}}_i; \bar{\mathbf{x}}_{<i})| = -\log|\det J_f(\bar{\mathbf{x}})|$.

When $m = 2$ and $f_1(\mathbf{x}_1) = \mathbf{x}_1$, the autoregressive flow is reduced to a *coupling flow*, which is widely used in flow-based models [10, 11, 16]. It is therefore trivially clear that the coupling flow is additionally compatible with Alg. 3.

### 2.3.2 $1 \times 1$ Convolutional Flow

$1 \times 1$ convolutional layers can be viewed as a matrix multiplication along a channel dimension [25]. Let $\bar{\mathbf{x}}, \bar{\mathbf{z}} \in \mathbb{R}^c$ be inputs and outputs along channels, and $\mathbf{W} \in \mathbb{R}^{c \times c}$ the weights of our network. The objective is to obtain $\bar{\mathbf{z}} = \bar{f}(\bar{\mathbf{x}})$ where $f(\mathbf{x}) = \mathbf{W}\mathbf{x}$.

We use the ideas of iVPF [41] to achieve a numerically invertible $1 \times 1$ convolutional transformation. In particular, we begin by performing an LU decomposition such that $\mathbf{W} = \mathbf{PL\Lambda U}$. It then follows that the $1 \times 1$ convolution is performed with successive matrix multiplications with $\mathbf{U}, \mathbf{\Lambda}, \mathbf{L}$ and $\mathbf{P}$. In iVPF, the authors extensively discussed matrix multiplications with factors $\mathbf{U}, \mathbf{L}$ and $\mathbf{P}$ [41]. Meanwhile, one can view the matrix multiplication with $\mathbf{\Lambda}$ as a scale transform, such that $f(\mathbf{x}) = \boldsymbol{\lambda} \odot \mathbf{x}$, where $\odot$ is an element-wise multiplication and $\boldsymbol{\lambda}$ are the diagonal elements of $\mathbf{\Lambda}$. MST in Alg. 1 can then be applied. In such a case, it is clear that Proposition 1 and 2 hold for a $1 \times 1$ convolutional flow. For Proposition 2, it is observed that $L_f(\bar{\mathbf{x}}, \bar{\mathbf{z}}) \approx -\text{sum}(\log \boldsymbol{\lambda}) = -\log|\det \mathbf{\Lambda}| = -\log|\det \mathbf{W}| = -\log|\det J_f(\bar{\mathbf{x}})|$.

## 2.4 Building Numerically Invertible Flows

Our flow model is constructed as a composition of layers $f = f_L \circ ... \circ f_2 \circ f_1$, where each layer is a transformation of the type discussed in Sec. 2.2 and 2.3. Let us represent the resulting flow as $\bar{f} = \bar{f}_L \circ ... \circ \bar{f}_1$, where $\bar{f}_l$ is a discretized transformation derived from the corresponding continuous $f$. It is clear that the quantized input $\bar{\mathbf{x}}(= \bar{\mathbf{y}}_0)$ and latent $\bar{\mathbf{z}}(= \bar{\mathbf{y}}_L)$ establish a bijection with successive transformations between discrete inputs and outputs. For the forward pass, $\bar{\mathbf{z}}$ is computed with $\bar{\mathbf{y}}_l = \bar{f}(\bar{\mathbf{y}}_{l-1}), l = 1, 2, ..., L$; for the inverse pass, $\bar{\mathbf{x}}$ is recovered with the inverse flow $\bar{f}^{-1} = \bar{f}_1^{-1} \circ ... \circ \bar{f}_L^{-1}$ such that $\bar{\mathbf{y}}_{l-1} = \bar{f}_l^{-1}(\bar{\mathbf{y}}_l)$.

For our resultant flow model, we can draw similar conclusions as in Propositions 1 and 2. Firstly, the error of $\bar{\mathbf{z}} = \bar{f}(\bar{\mathbf{x}})$ and $\mathbf{z} = f(\bar{\mathbf{x}})$ is small, bounded by $|\bar{\mathbf{z}} - \mathbf{z}| < O(LS^{-1}, L2^{-k}, L2^{-2h})$. Secondly, the codelength is approximately $L_f(\bar{\mathbf{x}}, \bar{\mathbf{z}}) = \sum_{l=1}^{L} L_{f_l}(\bar{\mathbf{y}}_{l-1}, \bar{\mathbf{y}}_l) \approx -\sum_{l=1}^{L} \log|\det J_{f_l}(\bar{\mathbf{y}}_{l-1})|$.

## 2.5 Lossless Compression with Flows via Bits-back Dequantization

Armed with our flow model $\bar{f}$, performing lossless compression is straight-forward. For the encoding process, the latent is generated according to $\bar{\mathbf{z}} = \bar{f}(\bar{\mathbf{x}})$. We then encode $\bar{\mathbf{z}}$ with probability $p_Z(\bar{\mathbf{z}})\delta$. For the decoding process, $\bar{\mathbf{z}}$ is decoded with $p_Z(\bar{\mathbf{z}})\delta$, and $\bar{\mathbf{x}}$ is recovered with $\bar{f}^{-1}(\bar{\mathbf{z}})$. The expected codelength is approximately $-\log(p_X(\bar{\mathbf{x}})\delta)$ such that

$$L(\bar{\mathbf{x}}) \approx -\log(p_Z(\bar{\mathbf{z}})\delta) - \sum_{l=1}^{L} \log|\det J_{f_l}(\bar{\mathbf{y}}_{l-1})| \approx -\log(p_X(\bar{\mathbf{x}})\delta). \tag{5}$$

However, we note that if $k$ is large, $-\log\delta = kd$ and the codelengths will also be large, resulting in a waste of bits. For what follows, we adopt the bits-back trick in LBB [17] to reduce the codelength. In particular, consider coding with input data $\mathbf{x}^\circ \in \mathbb{Z}^d$, where a $k$-precision noise vector $\bar{\mathbf{u}} \in [0,1)^d$ is decoded with $q(\bar{\mathbf{u}}|\mathbf{x}^\circ)\delta$ and added to input data such that $\bar{\mathbf{x}} = \mathbf{x}^\circ + \bar{\mathbf{u}}$. In this way, $\bar{\mathbf{x}}$ is then encoded with our flow $\bar{f}$. For the decoding process, $\mathbf{x}^\circ$ is recovered by applying the inverse transformation. We name this coding process *Bits-back Dequantization*, which is summarized in Alg. 4.

---

**Algorithm 4** Lossless Compression with iFlow.

---

**Encode $\mathbf{x}^\circ$.**
1: Decode $\bar{\mathbf{u}}$ using $q(\bar{\mathbf{u}}|\mathbf{x}^\circ)\delta$;
2: $\bar{\mathbf{z}} \leftarrow \bar{f}(\mathbf{x}^\circ + \bar{\mathbf{u}})$;
3: Encode $\bar{\mathbf{z}}$ using $p_Z(\bar{\mathbf{z}})\delta$.

**Decode.**
1: Decode $\bar{\mathbf{z}}$ using $p_Z(\bar{\mathbf{z}})\delta$;
2: $\bar{\mathbf{x}} \leftarrow \bar{f}^{-1}(\bar{\mathbf{z}}), \mathbf{x}^\circ \leftarrow \lfloor\bar{\mathbf{x}}\rfloor$;
3: Encode $\bar{\mathbf{u}} = \bar{\mathbf{x}} - \mathbf{x}^\circ$ using $q(\bar{\mathbf{u}}|\mathbf{x}^\circ)\delta$;
4: **return** $\mathbf{x}^\circ$.

---

In practice, $q(\mathbf{u}|\mathbf{x}^\circ)$ is constructed with a flow model such that $\mathbf{u} = g(\boldsymbol{\epsilon}; \mathbf{x}^\circ)$ where $\boldsymbol{\epsilon} \sim p(\boldsymbol{\epsilon})$. $\bar{\mathbf{u}}$ is decoded by first decoding $\bar{\boldsymbol{\epsilon}}$ with $p(\bar{\boldsymbol{\epsilon}})\delta$, and then applying $\bar{\mathbf{u}} = \bar{g}(\bar{\boldsymbol{\epsilon}}; \mathbf{x}^\circ)$. Thus decoding $\bar{\mathbf{u}}$ involves $-\log(q(\bar{\mathbf{u}}|\mathbf{x}^\circ)\delta)$ bits. Overall, the expected codelength is exactly the dequantization lower bound [19] such that

$$\begin{aligned} L(\mathbf{x}^\circ) &\approx \sum_{\bar{\mathbf{u}}} q(\bar{\mathbf{u}}|\mathbf{x}^\circ)\delta[\log(q(\bar{\mathbf{u}}|\mathbf{x}^\circ)\delta) - \log(p(\mathbf{x}^\circ + \bar{\mathbf{u}})\delta)] \\ &\approx \mathbb{E}_{q(\bar{\mathbf{u}}|\mathbf{x}^\circ)}[\log q(\bar{\mathbf{u}}|\mathbf{x}^\circ) - \log p(\mathbf{x}^\circ + \bar{\mathbf{u}})]. \end{aligned} \tag{6}$$

## 2.6 Extensions

With novel modifications, flow models can be applied to the generation of various data types, obtaining superior performance [7, 28, 23]. These models can be used for improved lossless compression. In general, given input data $\mathbf{x}^\circ$, these models generate intermediate data $\mathbf{v}$ with $q(\mathbf{v}|\mathbf{x}^\circ)$, and the density of $\mathbf{v}$ is modelled with flow model $\mathbf{z} = f(\mathbf{v})$. For generation, when $\mathbf{v}$ is generated with the inverse flow, $\mathbf{x}$ is generated with $p(\mathbf{x}^\circ|\mathbf{v})$. It is clear that $p(\mathbf{x})$ can be estimated with variational lower bound such that $\log p(\mathbf{x}^\circ) \geq \mathbb{E}_{q(\mathbf{v}|\mathbf{x}^\circ)}[\log P(\mathbf{x}^\circ|\mathbf{v}) + \log p(\mathbf{v}) - \log q(\mathbf{v}|\mathbf{x}^\circ)]$. For lossless compression, $\mathbf{x}^\circ$ can be coded with bits-back coding, which is similar with Alg. 4. In the encoding process, we first decode $\bar{\mathbf{v}}$ with $q(\bar{\mathbf{v}}|\mathbf{x}^\circ)\delta$ and then encode $\mathbf{x}^\circ$ with $P(\mathbf{x}^\circ|\bar{\mathbf{v}})$ (similar with Line 1 in Alg. 4-Encode). We then obtain the prior $\bar{\mathbf{z}} = \bar{f}(\bar{\mathbf{v}})$ (Line 2), before finally encoding $\bar{\mathbf{z}}$ with $p_Z(\bar{\mathbf{z}})\delta$ (Line 3). In the decoding process, $\bar{\mathbf{z}}$ is firstly decoded with $p_Z(\bar{\mathbf{z}})\delta$ (similar to Line 1 in Alg. 4-Decode), and then recovered $\bar{\mathbf{v}}$ and decoded $\mathbf{x}^\circ$ with $P(\mathbf{x}^\circ|\bar{\mathbf{v}})$ (Line 2). Finally, we encode using $\bar{\mathbf{v}}$ with $q(\bar{\mathbf{v}}|\mathbf{x}^\circ)$ [38]. The expected codelength is approximately

$$L(\mathbf{x}^\circ) \approx \mathbb{E}_{q(\bar{\mathbf{v}}|\mathbf{x}^\circ)}[\log q(\bar{\mathbf{v}}|\mathbf{x}^\circ) - \log P(\mathbf{x}^\circ|\bar{\mathbf{v}}) - \log p(\bar{\mathbf{v}})]. \tag{7}$$

We introduce a selection of recent, state-of-the-art flow-based models modified according to the above. Each model corresponds to a certain coding algorithm.

**VFlow [7].** VFlow expands the input data dimension with variational data augmentation to resolve the bottleneck problem in the flow model. In VFlow, $\mathbf{v} = [\mathbf{x}^\circ + \mathbf{u}, \mathbf{r}](\mathbf{u} \in [0,1)^d)$, where $\mathbf{u} \sim$

$q_u(\mathbf{u}|\mathbf{x}^\circ)$, $\mathbf{r} \sim q_r(\mathbf{r}|\mathbf{x}^\circ + \mathbf{u})$, is modelled with flows $g_u, g_r$ such that $\mathbf{u} = g_u(\boldsymbol{\epsilon}_u; \mathbf{x}^\circ)$, $\mathbf{r} = g_r(\boldsymbol{\epsilon}_r; \mathbf{x}^\circ + \mathbf{u})$ ($\boldsymbol{\epsilon}_u, \boldsymbol{\epsilon}_r$ are priors). Then we have $q(\mathbf{v}|\mathbf{x}^\circ) = q_u(\mathbf{u}|\mathbf{x}^\circ)q_r(\mathbf{r}|\mathbf{x}^\circ + \mathbf{u})$, $P(\mathbf{x}^\circ|\mathbf{v}) = 1$ (as $\mathbf{v} \to (\mathbf{x}^\circ + \mathbf{u}) \to \mathbf{x}^\circ$). Thus for the encoding process, $\bar{\boldsymbol{\epsilon}}_u, \bar{\boldsymbol{\epsilon}}_r$ are decoded. To construct $\bar{\mathbf{v}}$ we have $\bar{\mathbf{u}} = \bar{g}_u(\bar{\boldsymbol{\epsilon}}_u; \mathbf{x}^\circ), \bar{\mathbf{r}} = \bar{g}_r(\bar{\boldsymbol{\epsilon}}_r; \mathbf{x}^\circ + \bar{\mathbf{u}})$; and then $\bar{\mathbf{v}}$ is encoded with iFlow. For the decoding process, $\bar{\mathbf{v}}$ is decoded with the inverse iFlow, and then $\mathbf{x}^\circ, \bar{\mathbf{u}}, \bar{\mathbf{r}}$ is recovered with $\bar{\mathbf{v}}$. Here $\bar{\mathbf{u}}, \bar{\mathbf{r}}$ is encoded and $\mathbf{x}^\circ$ is the decoded output. As VFlow achieves better generation results compared with general flows, one would expect a better compression ratio with VFlow.

**Categorical Normalizing Flow [28].** Categorical Normalizing Flows (CNF) succeed in modelling categorical data such as text, graphs, etc. Given categorical data $\mathbf{x}^\circ = [x_1, ..., x_n], x_i \in \{1, 2, ..., C\}^n$, $\mathbf{v} = [\mathbf{v}_1, ..., \mathbf{v}_n]$ is represented with word embeddings such that $q(\mathbf{v}_i|x_i) = q_e(\mathbf{v}_i|\boldsymbol{\mu}(x_i), \boldsymbol{\sigma}(x_i))$, in which $q_e$ could be a Gaussian or logistic distribution. Then $P(x_i|\mathbf{v}_i) = \frac{\tilde{p}(x_i)q(\mathbf{v}_i|x_i)}{\sum_{c=1}^{C} \tilde{p}(c)q(\mathbf{v}_i|c)}$ with $\tilde{p}$ being the prior over categories. Thus $q(\mathbf{v}|\mathbf{x}^\circ) = \prod_{i=1}^{n} q(\mathbf{v}_i|x_i)$, $P(\mathbf{x}^\circ|\mathbf{v}) = \prod_{i=1}^{n} P(x_i|\mathbf{v}_i)$, and $\mathbf{x}^\circ$ can be coded with Alg. 8 given $q(\mathbf{v}|\mathbf{x}^\circ), P(\mathbf{x}^\circ|\mathbf{v})$ and the iFlow.

# 3 Uniform Base Conversion Systems

---

**Algorithm 5** Uniform Base Conversion Systems

---

| **ENCODE** $s$ **using** $U(0, R)(R < 2^K)$. | **DECODE with** $U(0, R)(R < 2^K)$. |
|---|---|
| **Input:** symbol $s$, state $c$, bit-stream $\mathtt{bs}$. | **Input:** state $c$, bit-stream $\mathtt{bs}$. |
| **Output:** new state $c$ and bit-stream $\mathtt{bs}$. | **Output:** decoded $s$, new state $c$ and bit-stream $\mathtt{bs}$. |

1: $c \leftarrow c \cdot R + s$;
2: **if** $c \geq 2^{M+K}$ **then**
3:     $\mathtt{bs.push\_back}(c \bmod 2^K)$;   ▷ push $K$ bits to bit-stream.
4:     $c \leftarrow \lfloor \frac{c}{2^K} \rfloor$;
5: **end if**
6: **return** $c, \mathtt{bs}$.

1: **if** $c < 2^M \cdot R$ **then**
2:     $c \leftarrow 2^K \cdot c + \mathtt{bs.pop\_back}()$;   ▷ get last $K$ bits from bit-stream and pop them.
3: **end if**
4: $s \leftarrow c \bmod R$;
5: $c \leftarrow \lfloor c/R \rfloor$;
6: **return** $s, c, \mathtt{bs}$.

---

The previous section demonstrates that coding with a uniform distribution is central to our algorithm. Note that the distribution varies in each coding process, thus *dynamic* entropy coder is expected. Compared to the Gaussian distribution used in LBB [17], a uniform distribution is simpler, yielding improved coding speed. As follows, we introduce our *Uniform Base Conversion Systems* (UBCS), which is easy to implement and the coding bandwidth is much greater than that of rANS [12].

UBCS is implemented based on a number-base conversion. The code state $c$ is represented as an integer. For coding some symbol $s \in \{0, 1, ..., R-1\}$ with a uniform distribution $U(0, R)$, the new state $c'$ is obtained by converting an $R$-base digit $s$ to an integer such that

$$c' = E(c, s) = c \cdot R + s. \tag{8}$$

For decoding with $U(0, R)$, given state $c'$, the symbol $s$ and state $c$ are recovered by converting the integer to an $R$-base digit such that

$$s = c' \mod R, \qquad c = D(c', s) = \lfloor \frac{c'}{R} \rfloor. \tag{9}$$

We note, however, that $c$ will become large when more symbols are encoded, and computing Eq. (8-9) with large $c$ will be inefficient. Similar to rANS, we define a "normalized interval" which bounds the state $c$ such that $c \in [2^M, 2^{K+M})$ ($K, M$ are some integers values) after coding each symbol. For the encoding process, if $c \geq 2^{K+M}$, the lower $K$ bits are written to disk, and the remaining bits are reserved such that $c \leftarrow \lfloor \frac{c}{2^K} \rfloor$. For the decoding process, if $c < 2^M \cdot R$, the stored $K$ bits should be read and appended to the state before decoding. In this way, the decoded state is contained within the interval $[2^M, 2^{K+M})$. The initial state can be set such that $c = 2^M$. We illustrate this idea in Alg. 5.

The correctness of Alg. 5 is illustrated in the following theorem. **P1** demonstrates that the symbols can be correctly decoded with a UBCS coder, with coding performed in a first-in-last-out (FILO) fashion. **P2** shows that the codelength closely approximates the entropy of a uniform distribution, subject to a large $M$ and $K$. The proof of the theorem is in the Appendix.

Table 2: Coding performance of iFlow, LBB and iVPF on CIFAR10 dataset. We use batch size 64.

| flow arch. | compression technique | nll | bpd | aux. bits | encoding time (ms) inference | coding | decoding time (ms) inference | coding |
|---|---|---|---|---|---|---|---|---|
| Flow++ | LBB [17] | 3.116 | **3.118** | 39.86 | $16.2_{\pm 0.3}$ | $116_{\pm 1.0}$ | $32.4_{\pm 0.2}$ | $112_{\pm 1.5}$ |
| | **iFlow (Ours)** | | **3.118** | **34.28** | | **$21.0_{\pm 0.5}$** | | **$37.7_{\pm 0.5}$** |
| iVPF | iVPF [41] | 3.195 | 3.201 | **6.00** | $5.5_{\pm 0.1}$ | $11.4_{\pm 0.2}$ | $5.2_{\pm 0.1}$ | $13.5_{\pm 0.3}$ |
| | **iFlow (Ours)** | | **3.196** | 7.00 | | **$7.1_{\pm 0.2}$** | | **$9.7_{\pm 0.2}$** |

**Theorem 3.** *Consider coding symbols $s_1, ...s_n$ with $s_i \sim U(0, R_i), (i = 1, 2, ..., n, R_i < 2^K)$ using Alg. 5, and then decode $s'_n, s'_{n-1}, ..., s'_1$ sequentially. Suppose (1) the initial state and bit-stream are $c_0 = 2^M, bs_0 = \texttt{empty}$ respectively; (2) After coding $s_i$, the state is $c_i$ and the bit-stream is $bs_i$; (3) After decoding $s'_{i+1}$, the state is $c'_i$ and the bit-stream is $bs'_i$. We have*

**P1:** $s'_i = s_{i+1}, c'_i = c_i, bs'_i = bs_i$ *for all $i = 0, ..., n - 1$.*

**P2:** *Denote by the codelength $l_i = \lceil \log c_i \rceil + \texttt{len}(bs_i)$ where $\texttt{len}$ is the total number of bits in the bit-stream. Then $l_n - l_0 < \frac{1}{1 - 1/(\ln 2 \cdot 2^M \cdot K)} \left[ \sum_{i=1}^{n} R_i + 1 + (\ln 2 \cdot 2^M)^{-1} \right]$.*

In practice, we set $K = 32$ and $M = 4$. Compared to rANS [12] and Arithmetic Coding (AC) [40], UBCS is of greater efficiency as it necessitates fewer operations (more discussions are shown in the Appendix). UBCS can achieve greater computational efficiency via instantiating multiple UBCS coders in parallel with multi-threading. Table 1 demonstrates that UBCS achieves coding bandwidths in excess of giga-symbol/s – speed significantly greater than rANS.

Table 1: Coding bandwidth (M symbol/s) of UBCS and rANS coder on different threads(thrd). We use the implementations in [17] for evaluating rANS.

| | # thrd | rANS | **UBCS** |
|---|---|---|---|
| Encoder | 1 | $5.1_{\pm 0.3}$ | **$380_{\pm 5}$** |
| | 16 | $21.6_{\pm 1.1}$ | **$2075_{\pm 353}$** |
| Decoder | 1 | $0.8_{\pm 0.02}$ | **$66.2_{\pm 1.7}$** |
| | 16 | $7.4_{\pm 0.5}$ | **$552_{\pm 50}$** |

## 4 Experiments

In this section, we perform a number of experiments to establish the effectiveness of iFlow. We will investigate: (1) how closely the codelength matches the theoretical bound; (2) the efficiency of iFlow as compared with the LBB [17] baseline; (3) the compression performance of iFlow on a series low and high-resolution images.

### 4.1 Flow Architectures and Datasets

We adopt two types of flow architectures for evaluation: Flow++ [16] and iVPF [41]. Flow++ is a state-of-the-art model using complex non-linear coupling layers and variational dequantizations [19]. iVPF is derived from a volume-preserving flow in which numerically invertible discrete-space operations are introduced. The models are re-implemented or directly taken from the corresponding authors. Unless specified, we use $h = 12, k = 28$ and set large $S$ – around $2^{16}$, which we analyse further in the Appendix. To reduce the auxiliary bits in the bits-back coding scheme, we partition the $d$-dimensional data into $b$ splits and perform MST (in Alg. 1 and 2) for each split sequentially. In this case, the auxiliary bits can be reduced to $1/b$ in MST. We use $b = 4$ in this experiment.

Following the lossless compression community [16, 4, 18, 37, 41], we perform evaluation using toy datasets CIFAR10, ImageNet32 and ImageNet64. Results for alternate methods are obtained via re-implementation or taken directly from the corresponding papers, where available. We further test the generalization capabilities of iFlow in which all toy datasets are compressed with a model trained on ImageNet32. For benchmarking our performance on high-resolution images, we evaluate iFlow using CLIC.mobile, CLIC.pro[2] and DIV2k [1]. For this purpose, we adopt our ImageNet32/64 model for evaluation, and process an image in terms of $32 \times 32$ or $64 \times 64$ patches, respectively. The experiment is conducted with PyTorch framework with one Tesla P100 GPU.

---

[2]https://www.compression.cc/challenge/

Table 3: Compression performance in bpd on benchmarking datasets. [†] denotes the generation performance in which the models are trained on ImageNet32 and tested on other datasets. [‡] denotes compression of high-resolution datasets with our ImageNet64-trained model.

| | ImageNet32 | ImageNet64 | CIFAR10 | CLIC.mobile | CLIC.pro | DIV2K |
|---|---|---|---|---|---|---|
| PNG [5] | 6.39 | 5.71 | 5.87 | 3.90 | 4.00 | 3.09 |
| FLIF [36] | 4.52 | 4.19 | 4.19 | 2.49 | 2.78 | 2.91 |
| JPEG-XL [2] | 6.39 | 5.74 | 5.89 | 2.36 | 2.63 | 2.79 |
| L3C [30] | 4.76 | 4.42 | - | 2.64 | 2.94 | 3.09 |
| RC [31] | - | - | - | 2.54 | 2.93 | 3.08 |
| Bit-Swap [26] | 4.50 | - | 3.82 | - | - | - |
| IDF [18] | 4.18 | 3.90 | 3.34 | - | - | - |
| IDF++ [4] | 4.12 | 3.81 | 3.26 | - | - | - |
| iVPF [41] | 4.03 | 3.75 | 3.20 | - | - | - |
| LBB [17] | **3.88** | **3.70** | **3.12** | - | - | - |
| **iFlow (Ours)** | **3.88** | **3.70** | **3.12** | - | - | - |
| HiLLoC [37][†] | 4.20 | 3.90 | 3.56 | - | - | - |
| IDF [18][†] | 4.18 | 3.94 | 3.60 | - | - | - |
| iVPF[†] [41] | 4.03 | 3.79 | 3.49 | 2.47/2.39[‡] | 2.63/2.54[‡] | 2.77/2.68[‡] |
| **iFlow (Ours)[†]** | **3.88** | **3.65** | **3.36** | **2.26/2.26[‡]** | **2.45/2.44[‡]** | **2.60/2.57[‡]** |

## 4.2 Compression Performance

For our experiments, we use the evaluation protocols of codelength and compression bandwidth. The codelength is defined in terms of the average bits per dimension (bpd). For no compression, the bpd is assumed to be 8. The compression bandwidth evaluates coding efficiency, which we define in terms of symbols compressed per unit time.

Table 2 demonstrates the compression results on CIFAR10. Note that we only report the encoding time (results on decoding time are similar, and are available in the Appendix). For iVPF, we use settings almost identical to the original paper [41] such that $k = 14$.

Firstly, we observe that, when using both the Flow++ and iVPF architectures, iFlow achieves a bpd very close to theoretically minimal codelength. When using Flow++, iFlow achieves identical performance as that of LBB. For the iVPF architecture, iFlow outperforms the underlying iVPF as it avoids the need to store 16 bits for each data sample.

Secondly, the coding latency highlights the main advantage of iFlow: we achieve encoding $5\times$ faster than that of LBB and over $1.5\times$ that of iVPF. In fact, the use of UBCS only represents $20\%$ of the total coding time for all symbols (4.8ms in Flow++ and 1.6ms in iVPF). In contrast, the rANS coder of LBB commands over $85\%$ of the total coding latency, which is the principal cause of LBB's impracticality. Indeed, Table 1 demonstrates that our UBCS coder achieves a speed-up in excess of $50\times$ that of rANS (which results in a coder latency of 4.8ms in iFlow vs. 99.8ms in LBB).

Lastly, compared with LBB, iFlow necessitates fewer auxiliary bits. In fact, LBB requires crica $2k + \log \sigma$ bits per dimension (for $\delta = 2^{-k}$ and small $\sigma$ in [17]). Meanwhile, iFlow requires approximately $k + \frac{1}{b} \log S$, and $\frac{1}{b} \log S$ is usually small with large $b$.

## 4.3 Comparison with the State-of-the-Art

To further demonstrate the effectiveness of iFlow, we compare the compression performance on benchmarking datasets against a variety of neural compression techniques. These include, L3C [30], Bit-swap [26], HilLoc [37], and flow-based models IDF [18], IDF++ [4], iVPF [41], LBB [17]. We additionally include a number of conventional methods, such as PNG [5], FLIF [36] and JPEG-XL [2].

**Experiments on Low Resolution Images.** Compression results on our described selection of datasets are available in left three columns of Table 3. Here we observe that iFlow obtains improved compression performance over all approaches with the exception of LBB on low-resolution images, for which we achieve identical results.

**Generalization.** The last four rows in Table 3 demonstrate the literature-standard test of generalization, in which ImageNet32 trained model are used for testing. From these results, it is clear that iFlow achieves the best generalization performance in this test. It is worth noting that we obtain an improved performance on ImageNet64 when using our ImageNet32-trained model.

**Experiments on High Resolution Images.** Finally, we test iFlow across a number of high-resolution image datasets. Here images are processed into non-overlapping $32 \times 32$ and $64 \times 64$ patches for our ImageNet32 and ImageNet64-trained models. The right three columns in Table 3 display the results, which is observed that iFlow outperforms all compression methods across all available benchmarks. Note that as we crop patches for compression, the compression bandwidth is the same as that in small images like CIFAR10, i.e., 5 times faster than LBB and 30% speedup compared with iVPF.

## 5 Related Work

*Dynamic* Entropy coders, such as Arithmetic Coding (AC) [40] and Asymmetric Numerical Systems (ANS) [12], form the basis of lossless compression. However, the binary search protocol required at decode time and their comparatively high number of numerical operations make them both relatively time-consuming. The term *dynamic* means that the data symbols are in different distributions, in this case, efficient entropy coders like Huffman coding [21] and tANS [12] are incompatible. Our proposed UBCS is *dynamic* coder, which requires only two operations per symbol, producing an faster algorithm than AC and ANS.

In order to utilise entropy coders, one must estimate the data distribution. For this purpose, the wider community has employed a variety of density estimators. One of the most popular, autoregressive models [35], estimates the joint density with per-pixel autoregressive factorizations. Whilst commonly achieving state-of-the-art compression ratios, the sequential pixel-by-pixel nature of encoding and/or decoding makes them impractically time-consuming. Alternatively, variational autoencoders (VAEs) [24, 26, 37] maximise a lower bound on the marginal data likelihood (otherwise known as the *ELBO*). With the bits-back coding framework [38], the theoretical codelength is exactly equal to the ELBO. However, in most cases, VAE formulations typically produce inferior compression ratios as there exists a gap between the ELBO and the true data likelihood.

As discussed, flow-based models [16, 25, 11], which admit exact likelihood computation, represent an alternative route for density estimation. IDF [18] and IDF++ [4] proposed the titular *integer discrete flow* to preserve the existence and uniqueness of an invertible mapping between discrete data and latent variables. In a similar vein, iVPF [41] achieved a mapping with volume-preserving flows. Here the remainders of a division operation are stored as auxiliary states to eliminate the numerical error arising from discretizing latent variables. However, all of these models must introduce constraints on the underlying transform, limiting their representational power. LBB [17] was the first flow-based lossless compression approach to admit a broad class of invertible flows based on *continuous* transforms. LBB established this family of flexible bijections by introducing local bits-back coding techniques to encode numerical errors. However, LBB typical requires the coding of many such errors and does so with the ANS scheme, posing obvious challenges to computational efficiency.

## 6 Conclusions and Discussions

In this paper, we have proposed iFlow, a numerically invertible flow-based model for achieving efficient lossless compression with state-of-the-art compression ratios. To achieve this, we have introduced the Modular Scale Transform and Uniform Base Conversion Systems, which jointly permit an efficient bijection between discrete data and latent variables. Experiments demonstrate that the codelength comes extremely close to the theoretically minimal value, with compression achieved much faster than the next-best high-performance scheme. Moreover, iFlow is able to achieve state-of-the-art compression ratios on real-word image benchmarks.

We additionally consider the potential for extending iFlow. That is, recent advances in normalizing flows have achieved improved generation performance across various data types [28, 7, 23]. We have discussed the possible extension to incorporate these advancements in Sec. 2.6, and consider its application as future work. We further recognise that compression aproaches, of which iFlow is one, present significant data privacy issues. That is, the generative model used for compression may induce data leakage; therefore, the codec should be treated carefully as to observe data-privacy laws.

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
