# A  Proofs

## A.1  Correctness of MST (Alg. 1)

As $\bar{x}, \bar{z}$ are quantized to $k$-precision, $\hat{x}$ and $\bar{x}$ form a bijection, and so do $\hat{z}$ and $\bar{z}$. Thus MST is correct if and only if $\hat{x}$ and $\hat{z}$ are valid bijections. In particular, denoting $\hat{x}^d = \lfloor (S \cdot \hat{z} + r_e)/R \rfloor, r_d^d = (S \cdot \hat{z} + r_e) \mod R$, we will show $\hat{x}^d \equiv \hat{x}, r_d^d \equiv r_d$.

In fact, according to forward MST, $\hat{z} = \lfloor (R \cdot \hat{x} + r_d)/S \rfloor, r_e = (R \cdot \hat{x} + r_d) \mod S$, thus $S \cdot \hat{z} + r_e = R \cdot \hat{x} + r_d$. Considering $r_e$ is first accurately decoded by inverse MST (in Line 2), it is clear that $\hat{x}^d = \lfloor (S \cdot \hat{z} + r_e)/R \rfloor = \lfloor (R \cdot \hat{x} + r_d)/R \rfloor = \hat{x}$, and $r_d^d = (S \cdot \hat{z} + r_e) \mod R = (R \cdot \hat{x} + r_d)$ mod $R = r_d$. Thus the correctness of MST is proven.

## A.2  Propositions 1-2 in MST (Alg. 1)

Firstly, as $S \cdot a - 0.5 \le R < S \cdot a + 0.5$, we have $|a - \frac{R}{S}| \le \frac{0.5}{S} = O(S^{-1})$. Secondly, as $\bar{z} = \lfloor (R \cdot 2^k \cdot \bar{x} + r_d)/S \rfloor/2^k$ where $r_d \in [0, R)$, we have $\bar{z} < (R \cdot 2^k \cdot \bar{x} + R)/(S \cdot 2^k) = R/S \cdot \bar{x} + (R-1)/(S \cdot 2^k)$ and $\bar{z} > (R \cdot 2^k \cdot \bar{x}/S - 1)/2^k = R/S \cdot \bar{x} - 2^{-k}$, thus $|\bar{z} - R/S \cdot \bar{x}| < O(2^{-k})$. Then the error between $\bar{z}$ and $f(\bar{x})$ is $|\bar{z} - a \cdot \bar{x}| \le |\bar{z} - R/S \cdot \bar{x}| + |(a - R/S) \cdot \bar{x}| = O(S^{-1}, 2^{-k})$; therefore, Proposition 1 holds.

For Proposition 2, decoding from $U(0, R)$ involves $-\log R$ bits, and encoding $U(0, S)$ involves $\log S$ bits. As $|a - R/S| < O(S^{-1})$ and $f'(\bar{x}) = a$, the codelength of MST is $L_f(\bar{x}, \bar{z}) = \log S - \log R = -\log |a + O(S^{-1})| = -\log |f'(\bar{x})| + O(S^{-1}) \approx -\log |f'(\bar{x})|$. Thus Proposition 2 holds.

## A.3  Correctness of our Invertible Non-linear Flows (Alg. 2)

If the interpolation interval $[\bar{x}_l, \bar{x}_h)$ and $[\bar{z}_l, \bar{z}_h)(\bar{z}_l = \overline{f(\bar{x}_l)}, \bar{z}_l = \overline{f(\bar{x}_h)})$ are identical in both the forward and inverse processes, $f_{inp}$ is additionally identical in both the forward and inverse processes. Consequently, an exact bijection with MST algorithm is trivially achieved. Thus we principally seek to show that the interpolation interval can be correctly determined. Before the proof, it must be emphasised that the interpolation interval should demonstrate the following properties:

1. The interpolation interval is counted and covers the domain/co-domain. $\bar{x}_l, \bar{x}_h$ must be within the discretized set such that $\bar{x}_l, \bar{x}_h \in \mathcal{X}_{inp}$ (e.g. $\mathcal{X}_{inp} = \{2^{-h} \cdot n, n \in \mathbb{Z}\}$);

2. The interpolation interval must be not intersected. There does not exist $x$ such that $x \in [\bar{x}_l, \bar{x}_h)$ and $x \in \mathcal{X}_{inp} \setminus \{\bar{x}_l\}$.

Firstly, we show that in the forward computation, $\bar{z}$ will always be within $[\bar{z}_l, \bar{z}_h)$. In fact, $\bar{z} = \lfloor \frac{R \cdot 2^k (\bar{x} - \bar{x}_l) + r_d}{S} \rfloor/2^k + \bar{z}_l$ where $r_d \in [0, R)$. As $\bar{x} \ge \bar{x}_l, \bar{z} \ge \lfloor \frac{R \cdot 0 + 0}{S} \rfloor/2^k + \bar{z}_l = \bar{z}_l$. As $\bar{x} \le \bar{x}_h - 2^{-k}$, $\bar{z} \le \lfloor \frac{R \cdot 2^k (\bar{x}_h - \bar{x}_l - 2^{-k}) + R - 1}{S} \rfloor/2^k + \bar{z}_l$, following from Eq. (4) it is clear that $\bar{z} \le \bar{z}_h - 2^{-k}$. Therefore, it holds that $\bar{z} \in [\bar{z}_l, \bar{z}_h)$.

Secondly, we show that during the inverse computation, the interpolation intervals $[\bar{x}_l, \bar{x}_h)$ and $[\bar{z}_l, \bar{z}_h)$ are the same as that in the forward process. In other words, if the interpolation interval is $[\bar{x}_l^d, \bar{x}_h^d), [\bar{z}_l^d, \bar{z}_h^d)$ given $\bar{z}$, it holds that $\bar{x}_l^d \equiv \bar{x}_l, \bar{x}_h^d \equiv \bar{x}_h$. In fact, $\bar{x}_l^d \ge \bar{x}_h$ is not preserved, as $\overline{f(\bar{x}_l^d)} \ge \bar{z}$. It then follows that $\bar{z}$ could not be in $[\bar{z}_l^d, \bar{z}_h^d)$. Similarly, $\bar{x}_h \le \bar{x}_l$ is not preserved – otherwise $\overline{f(\bar{x}_h^d)} \le \bar{z}$ and $\bar{z}$ could not be in $[\bar{z}_l^d, \bar{z}_h^d)$. Overall, $\bar{x}_l^d < \bar{x}_h$ and $\bar{x}_h^d > \bar{x}_l$, such that only $\bar{x}_l^d = \bar{x}_l, \bar{x}_h^d = \bar{x}_h$ satisfy this condition.

## A.4  Propositions 1-2 in Invertible Non-linear Flows (Alg. 2)

In general, $f'(x), f''(x)$ is bounded for all $x$. Consider $f(\bar{x})$ in the small interval such that $\bar{x}_h - \bar{x}_l \le 2^{-h}$ or $\bar{z}_h - \bar{z}_l \le 2^{-h}$ ($k \gg h$). We first prove the following two propositions in $[\bar{x}_l, \bar{x}_h)$:

1. $|f'(\bar{x}) - \frac{\bar{z}_h - \bar{z}_l}{\bar{x}_h - \bar{x}_l}| < O(2^{-h}, 2^{h-k})$;

2. $|f_{inp}(\bar{x}) - f(\bar{x})| < O(2^{-2h}, 2^{-k})$

In fact, by performing the Taylor expansion at $\bar{x}$, we have $f(\bar{x}_l) = f(\bar{x}) + (\bar{x}_l - \bar{x}) \cdot f'(\bar{x}) + \frac{(\bar{x}_l - \bar{x})^2}{2} \cdot f''(\xi_l)$ and $f(\bar{x}_h) = f(\bar{x}) + (\bar{x}_h - \bar{x}) \cdot f'(\bar{x}) + \frac{(\bar{x}_h - \bar{x})^2}{2} \cdot f''(\xi_h)$, where $\xi_h, \xi_l \in (\bar{x}_l, \bar{x}_h)$.

Firstly, $f(\bar{x}_h) - f(\bar{x}_l) = (\bar{x}_h - \bar{x}_l) \cdot f'(x) + [\frac{(\bar{x}_h - \bar{x})^2}{2} \cdot f''(\xi_h) - \frac{(\bar{x}_l - \bar{x})^2}{2} \cdot f''(\xi_l)]$. As $f'(x), f''(x)$ are bounded, $\bar{x}_h - \bar{x}_l \le O(2^{-h})$, and we have $f(\bar{x}_h) - f(\bar{x}_l) = (\bar{x}_h - \bar{x}_l) \cdot f'(x) + O(2^{-2h})$. As $|\bar{z}_l - f(\bar{x}_l)| < 2^{-k}, |\bar{z}_l - f(\bar{x}_l)| < 2^{-k}$, then $|\bar{z}_h - \bar{z}_l - (\bar{x}_h - \bar{x}_l) \cdot f'(x)| < O(2^{-k}, 2^{-2h})$. Finally, $|f'(\bar{x}) - \frac{\bar{z}_h - \bar{z}_l}{\bar{x}_h - \bar{x}_l}| < O(2^{-h}, 2^{h-k})$.

Secondly, by denoting $g(\bar{x}) = \frac{f(\bar{x}_h) - f(\bar{x}_l)}{\bar{x}_h - \bar{x}_l}(\bar{x} - \bar{x}_l) + f(\bar{x}_l)$ and replacing $f(\bar{x}_l), f(\bar{x}_h)$ with its Taylor expansion, we have $g(\bar{x}) = f(\bar{x}) + \frac{1}{2(\bar{x}_h - \bar{x}_l)}(\bar{x}_h - \bar{x})(\bar{x} - \bar{x}_l)[(\bar{x}_h - \bar{x})f''(\xi_h) + (\bar{x} - \bar{x}_l)f''(\xi_l)]$. As $f''(x)$ is bounded, $|g(\bar{x}) - f(\bar{x})| < O(2^{-2h})$. Moreover, it is clear that $|f_{inp}(\bar{x}) - g(\bar{x})| < O(2^{-k})$, and as such it finally holds that $|f_{inp}(\bar{x}) - f(\bar{x})| < O(2^{-2h}, 2^{-k})$.

For Proposition 1, with MST, we have $\bar{z} - \bar{z}_l = \lfloor \frac{R \cdot 2^k \cdot (\bar{x} - \bar{x}_l) + r_d}{S} \rfloor / 2^k$ where $r_d \in [0, R)$, thus $|\bar{z} - \bar{z}_l - \frac{R}{S} \cdot (\bar{x} - \bar{x}_l)| < O(2^{-k})$. Moreover, with Eq. (4), it is easy to arrive at that $\frac{\bar{z}_h - \bar{z}_l - 2^{-k}(1 - S^{-1})}{\bar{x}_h - \bar{x}_l} - S^{-1} < \frac{R}{S} \le \frac{\bar{z}_h - \bar{z}_l - 2^{-k}(1 - S^{-1})}{\bar{x}_h - \bar{x}_l}$, and therefore $|\frac{R}{S} - \frac{\bar{z}_h - \bar{z}_l}{\bar{x}_h - \bar{x}_l}| < O(S^{-1}, 2^{h-k})$. Overall, $|\bar{z} - f_{inp}(\bar{x})| = |(\bar{z} - \bar{z}_l - \frac{R}{S} \cdot (\bar{x} - \bar{x}_l)) + ((\frac{R}{S} - \frac{\bar{z}_h - \bar{z}_l}{\bar{x}_h - \bar{x}_l}) \cdot (\bar{x} - \bar{x}_l))| < O(S^{-1}, 2^{-k})$, and finally $|\bar{z} - f(\bar{x})| \le |\bar{z} - f_{inp}(\bar{x})| + |f_{inp}(\bar{x}) - f(\bar{x})| < O(S^{-1}, 2^{-k}, 2^{-2h})$.

For Proposition 2, with $|\frac{R}{S} - \frac{\bar{z}_h - \bar{z}_l}{\bar{x}_h - \bar{x}_l}| < O(S^{-1}, 2^{h-k})$, it is clear that the expected codelength is $L_f(\bar{x}, \bar{z}) = -\log(R/S) = -\log \frac{\bar{z}_h - \bar{z}_l}{\bar{x}_h - \bar{x}_l} + O(S^{-1}, 2^{h-k}) = -\log|f'(\bar{x})| + O(S^{-1}, 2^{h-k}, 2^{-h})$. Overall, $L_f(\bar{x}, \bar{z}) \approx -\log|f'(\bar{x})|$ if $S, k, h$ are large and $k \gg h$.

## A.5 Theorem 3 in UBCS (Alg. 5)

**P1**. We begin by showing that $c_i \in [2^M, 2^{K+M})$ for all $i = 0, ..., n$ with mathematical induction. In fact, when $i = 0$, $c_0 = 2^M \in [2^M, 2^{K+M})$. When $i = k$ and $c_k \in [2^M, 2^{K+M})$, denote $c_{k+1}^\circ = c_k \cdot R_{k+1} + s_{k+1}$. It is therefore clear that $c_{k+1}^\circ = [2^M \cdot R_{k+1}, 2^{M+K} \cdot R_{k+1})$. Note that $R_{k+1} < 2^K$ and therefore $2^M \cdot R_{k+1} < 2^{M+K}$. If $c_{k+1}^\circ < 2^{K+M}$, $c_{k+1} = c_{k+1}^\circ \in [2^M \cdot R_{k+1}, 2^{M+K})$, it follows that $c_{k+1}^\circ \ge 2^{K+M}$, $c_{k+1} = \lfloor c_{k+1}^\circ / 2^K \rfloor \in [2^M, 2^K \cdot R_{k+1})$. Overall, $c_{k+1} \in [2^M, 2^{K+M})$. Thus $c_i \in [2^M, 2^{K+M})$ for all $i = 1, ..., n$ such that

$$c_i = \begin{cases} c_{i-1} \cdot R_i + s_i \in [2^M \cdot R_i, 2^{K+M}), & c_{i-1} \cdot R_i + s_i < 2^{K+M} \\ \lfloor \frac{c_{i-1} \cdot R_i + s_i}{2^K} \rfloor \in [2^M, 2^M \cdot R_i), & c_{i-1} \cdot R_i + s_i \ge 2^{K+M} \end{cases} \quad (10)$$

We will now demonstrate that $s_i' = s_{i+1}, c_i' = c_i, \text{bs}_i' = \text{bs}_i$ for all $i = 0, ..., n - 1$. Denote $c_i^\circ = c_{i-1} \cdot R_i + s_i$.

(i) Consider $i = n - 1$. (a) If $c_n < 2^M \cdot R_n$, the last $K$ bits (denoted by $r_n$) will be popped from $\text{bs}_n$ and added to $c_n$. In this case, according to Eq. (10), $r_n = \lfloor c_n^\circ \mod 2^K \rfloor$ must be encoded to form $\text{bs}_n$. Thus in the decoding process, $r_n$ is popped from $\text{bs}_n$, and therefore $\text{bs}_{n-1}' = \text{bs}_{n-1}$, $c_{n-1}' = \lfloor (2^K \cdot c_n + r_n)/R_n \rfloor = \lfloor c_{n-1}^\circ / R_n \rfloor = \lfloor (c_{n-1} \cdot R_n + s_{n-1})/R_n \rfloor = c_{n-1}$, and $s_{n-1}' = (2^K \cdot c_n + r_n) \mod R_n = (c_{n-1} \cdot R_n + s_{n-1}) \mod R_n = s_{n-1}$. (b) If $c_n \ge 2^M \cdot R$, no bits are popped from $\text{bs}_n$ such that $\text{bs}_{n-1}' = \text{bs}_n$. In this case, according to Eq. (10), no bits are pushed to $\text{bs}_{n-1}$ and therefore $\text{bs}_{n-1} = \text{bs}_n = \text{bs}_{n-1}'$. In the decoding process, $c_{n-1}' = \lfloor c_{n-1}^\circ / R_n \rfloor = c_{n-1}$, $s_{n-1}' = c_{n-1}^\circ \mod R_n = s_{n-1}'$. Overall, **P1** holds for $i = n - 1$.

(ii) If **P1** holds for $i = k$, we will prove that **P1** holds for $i = k - 1$. (a) If $c_k' < 2^M \cdot R_k$, the last $K$ bits will be popped from $\text{bs}_k'$ and added to $c_k'$. In this case, in the encoding process, as $c_k = c_k'$, according to Eq. (10), $r_k = \lfloor c_k^\circ \mod 2^K \rfloor$ must be encoded to form $\text{bs}_k$ to obtain $c_k$. In the decoding process, as $\text{bs}_k' = \text{bs}_k$, $r_k$ is popped from $\text{bs}_k'$ in the decoding process, it is therefore seen that $\text{bs}_{k-1}' = \text{bs}_{k-1}$. Finally we obtain $c_{k-1}' = \lfloor (2^K \cdot c_k' + r_k)/R_k \rfloor = \lfloor (2^K \cdot c_k + r_k)/R_k \rfloor = \lfloor c_{k-1}^\circ / R_k \rfloor = c_{k-1}$, and $s_{k-1}' = (2^K \cdot c_k' + r_k) \mod R_k = (c_{k-1} \cdot R_n + s_{k-1}) \mod R_n = s_{k-1}$. (b) If $c_k' \ge 2^M \cdot R$, no bits are popped from $\text{bs}_k'$ such that $\text{bs}_{k-1}' = \text{bs}_k'$. In this case, in the encoding process, as $c_k' = c_k$, according to Eq. (10), no bits are pushed to

$\text{bs}_{k-1}$ to obtain $c_k$ and therefore $\text{bs}_{k-1} = \text{bs}_k = \text{bs}'_k = \text{bs}'_{k-1}$. In the decoding process, we have $c'_{k-1} = \lfloor (c'_{k-1} \cdot R_k + s_{k-1})/R_k \rfloor = \lfloor c^o_{k-1}/R_k \rfloor = c_{k-1}, s'_{k-1} = (c'_{k-1} \cdot R_k + s_{k-1}) \bmod R_k = s_{k-1}$. Overall, **P1** holds for $i = k - 1$.

From (i)(ii), it is concluded that **P1** holds by proof of mathematical induction.

**P2**. Denote that the lower $K$ bits of $c_i$ need to be push to $\text{bs}_i$ at $i = m_1, ..., m_T$ ($m_t < m_{t+1}, m_t \in \{1, ..., n-1\}$ for all $t = 1, ..., T - 1$). In other words, we have

$$c_i = \begin{cases} \lfloor \frac{c_{i-1} \cdot R_i + s_i}{2^K} \rfloor, & i \in \{m_1, ..., m_T\} \\ c_{i-1} \cdot R_i + s_i, & otherwise \end{cases} \tag{11}$$

Firstly, it is clear that $\text{len}(\text{bs}_{m_{t+1}}) = \text{len}(\text{bs}_{m_t}) + K$. Secondly, for any $i \in \{m_t+1, ..., m_{t+1}-1\}$, as $c_i = c_{i-1} \cdot R_i + s_i, s_i \in [0, R_i)$, it is clear that $c_{i-1} \cdot R_i \le c_i \le c_{i-1} \cdot R_i + R_i - 1$. Thus $c_{m_t} \cdot \prod_{i=m_t+1}^{m_{t+1}-1} R_i \le c_{m_{t+1}-1} \le (c_{m_t} + 1) \cdot \prod_{i=m_t+1}^{m_{t+1}-1} R_i - 1$, and therefore

$$\lfloor \frac{c_{m_t} \cdot \prod_{i=m_t+1}^{m_{t+1}} R_i}{2^K} \rfloor \le c_{m_{t+1}} \le \lfloor \frac{(c_{m_t} + 1) \cdot \prod_{i=m_t+1}^{m_{t+1}} R_i - 1}{2^K} \rfloor. \tag{12}$$

Note that the above inequality also holds for $t = 0$ in which $m_0 = 0$. With Eq. (12), $\log c_{m_{t+1}} \le \log \lfloor \frac{(c_{m_t}+1)\cdot \prod_{i=m_t+1}^{m_{t+1}} R_i - 1}{2^K} \rfloor < \log \left( \frac{(c_{m_t}+1)\cdot \prod_{i=m_t+1}^{m_{t+1}} R_i}{2^K} \right) = \log c_{m_t} + \sum_{i=m_t+1}^{m_{t+1}} \log R_i + \log(1 + c_{m_t}^{-1}) - K < \log c_{m_t} + \sum_{i=m_t+1}^{m_{t+1}} \log R_i + (\ln 2 \cdot c_{m_t})^{-1} - K$. As $c_{m_t} \in [2^M, 2^M \cdot R_{m_t})$, it holds that

$$\log c_{m_{t+1}} + \text{len}(\text{bs}_{m_{t+1}}) < \log c_{m_t} + \text{len}(\text{bs}_{m_t}) + \sum_{i=m_i+1}^{m_{t+1}} \log R_i + (\ln 2 \cdot 2^M)^{-1} \tag{13}$$

If $m_T = n$, $\log c_n + \text{len}(\text{bs}_n) = \log c_{m_T} + \text{len}(\text{bs}_{m_T})$; otherwise, $\text{len}(\text{bs}_n) = \text{len}(\text{bs}_{m_T})$ and $\log c_n \le \log(c_{m_T} + 1) + \sum_{i=m_T+1}^{n} R_i < \log c_{m_T} + \sum_{i=m_T+1}^{n} R_i + (\ln 2 \cdot 2^M)^{-1}$. Overall, we finally obtain

$$\log c_n + \text{len}(\text{bs}_n) < \log c_0 + \text{len}(\text{bs}_0) + \sum_{i=1}^{n} \log R_i + (T + 1) \cdot (\ln 2 \cdot 2^M)^{-1} \tag{14}$$

Note that as $c_n \ge 2^M = c_0$, $\text{len}(\text{bs}_n) = TK$ and $l_0 = M = \log c_0 + \text{len}(\text{bs}_0)$, it is clear that $\log c_n + \text{len}(\text{bs}_n) > TK + l_0$. With Eq. (14), the codelength is finally computed as

$$l_n - l_0 \le \log c_n + \text{len}(\text{bs}_n) - l_0 + 1 < \frac{1}{1 - 1/(\ln 2 \cdot 2^M \cdot K)} \Big[ \sum_{i=1}^{n} R_i + 1 + (\ln 2 \cdot 2^M)^{-1} \Big] \tag{15}$$

which completes the proof.

# B  Details of Alg. 2 in iFlow

The main difficulty is in determining the interpolation interval $[\bar{x}_l, \bar{x}_h), [\bar{z}_l, \bar{z}_h)$ given $\bar{x}$ or $\bar{z}$. The main paper discusses two interpolation tricks: (1) interpolating uniform intervals in domain $x$ and (2) interpolating uniform intervals in co-domain $z$.

**Interpolating uniform intervals in $x$.** This usually applies in the case that $f'$ is large (e.g. inverse sigmoid). The uniform interval in domain $x$ is defined as $\bar{x}_l = \lfloor (2^h \cdot \bar{x})/2^h \rfloor, \bar{x}_h = \bar{x}_l + 2^{-h}$. The corresponding interval in the co-domain is $\bar{z}_l = \overline{f(\bar{x}_l)}, \bar{z}_h = \overline{f(\bar{x}_h)}$. For the forward pass, given $\bar{x}$, the interval can be obtained as above. For the inverse pass, we first compute $x' = f^{-1}(\bar{z})$ and then compute $\bar{x}'_m = \text{round}(2^h \cdot x')/2^h, \bar{x}'_l = \bar{x}'_m - 2^{-h}, \bar{x}'_h = \bar{x}'_m + 2^{-h}$. Finally, we have $\bar{z}_{\{l,m,h\}} = \overline{f(\bar{x}'_{\{l,m,h\}})}$. If $\bar{z} < \bar{z}_m$, we set the interval $\bar{x}_l = \bar{x}'_l, \bar{x}_h = \bar{x}'_m$; otherwise $\bar{x}_l = \bar{x}'_m, \bar{x}_h = \bar{x}'_h$. The method is summarized in Alg. 6.

The correctness of the algorithm is guaranteed provided that $f$ is Lipchitz continuous and the numerical error between $f^{-1}(f(\bar{x}))$ and $x$ is limited. Let $|f^{-1}(f(x)) - x| < \epsilon$ for all $x$ and

**Algorithm 6** Uniform interpolating interval $x$ in numerically invertible element-wise flows.

**Determine** $\bar{x}_l, \bar{x}_h, \bar{z}_l, \bar{z}_h$ **given** $\bar{x}$.
1: $\bar{x}_l = \lfloor (2^h \cdot \bar{x})/2^h \rfloor, \bar{x}_h = \bar{x}_l + 2^{-h}$;
2: $\bar{z}_l = \overline{f(\bar{x}_l)}, \bar{z}_h = \overline{f(\bar{x}_h)}$;
3: **return** $\bar{x}_l, \bar{x}_h, \bar{z}_l, \bar{z}_h$.

**Determine** $\bar{x}_l, \bar{x}_h, \bar{z}_l, \bar{z}_h$ **given** $\bar{z}$.
1: $x' = f^{-1}(\bar{z})$;

2: $\bar{x}'_m = \text{round}(2^h \cdot x')/2^h, \bar{x}'_l = \bar{x}'_m - 2^{-h}, \bar{x}'_h = \bar{x}'_m + 2^{-h}$;
3: $\bar{z}'_{\{l,m,h\}} = \overline{f(\bar{x}'_{\{l,m,h\}})}$;
4: **if** $\bar{z} < \bar{z}'_m$ **then**
5: $\quad \bar{x}_l = \bar{x}'_l, \bar{x}_h = \bar{x}'_m, \bar{z}_l = \bar{z}'_l, \bar{z}_h = \bar{z}'_m$;
6: **else**
7: $\quad \bar{x}_l = \bar{x}'_m, \bar{x}_h = \bar{x}'_h, \bar{z}_l = \bar{z}'_m, \bar{z}_h = \bar{z}'_h$
8: **end if**
9: **return** $\bar{x}_l, \bar{x}_h, \bar{z}_l, \bar{z}_h$.

$|f^{-1}(x_1) - f^{-1}(x_2)| < \mu |x_1 - x_2|$ for all $x_1, x_2$. We will show that Alg. 6 is correct if $\varepsilon = 2^{-k}\mu + \epsilon < 2^{-h-1}$. In fact, it is clear that $|f^{-1}(\bar{z}_l) - \bar{x}_l| \leq |f^{-1}(\bar{z}_l) - f^{-1}(f(\bar{x}_l))| + |\bar{x}_l - f^{-1}(f(\bar{x}_l))| < 2^{-k}\mu + \epsilon = \varepsilon$. Similarly, $|f^{-1}(\bar{z}_h) - \bar{x}_h| < \varepsilon$. As $f^{-1}$ is monotonically increasing and $\bar{z} \in (\bar{z}_l, \bar{z}_h)$, it is clear that $x' = f^{-1}(\bar{z}) \in [\bar{x}_l - \varepsilon, \bar{x}_h + \varepsilon)$. As $|\bar{x}'_m - x'| \leq 2^{-h-1}$, we have $\bar{x}'_m \in (\bar{x}_l - \varepsilon - 2^{-h-1}, \bar{x}_h + \varepsilon + 2^{-h-1})$. (i) When $\bar{z} < \bar{z}'_m$, it corresponds to $\bar{x} < \bar{x}'_m$. As $\bar{x} \in [\bar{x}_l, \bar{x}_h)$, it is clear that $\bar{x}'_m \in (\bar{x}_l, \bar{x}_h + \varepsilon + 2^{-h-1}) = (\bar{x}_h - 2^{-h}, \bar{x}_h + \varepsilon + 2^{-h-1})$. Note that $\bar{x}'_m, \bar{x}_h \in \{2^{-h} \cdot n, n \in \mathbb{Z}\}$ when $\varepsilon < 2^{-h-1}$; therefore, it must hold that $\bar{x}'_m = \bar{x}_h$. (ii) When $\bar{z} \geq \bar{z}'_m$, it corresponds to $\bar{x} \geq \bar{x}'_m$. It is clear that $\bar{x}'_m \in (\bar{x}_l - \varepsilon - 2^{-h-1}, \bar{x}_h) = (\bar{x}_l - \varepsilon - 2^{-h-1}, \bar{x}_l + 2^{-h})$ – thus it must hold that $\bar{x}'_m = \bar{x}_l$. In fact, as $h \ll k$, $\mu$ is bounded and $\epsilon$ is rather small, such that $\varepsilon < 2^{-h-1}$, the correctness of Alg. 6 follows.

---

**Algorithm 7** Uniform interpolating interval $z$ in numerically invertible element-wise flows.

**Determine** $\bar{x}_l, \bar{x}_h, \bar{z}_l, \bar{z}_h$ **given** $\bar{x}$.
1: $z = f(\bar{z})$;
2: $\bar{z}'_m = \text{round}(2^h \cdot z')/2^h, \bar{z}'_l = \bar{z}'_m - 2^{-h}, \bar{z}'_h = \bar{z}'_m + 2^{-h}$;
3: $\bar{x}'_{\{l,m,h\}} = \overline{f^{-1}(\bar{z}'_{\{l,m,h\}})}$;
4: **if** $\bar{x} < \bar{x}'_m$ **then**
5: $\quad \bar{x}_l = \bar{x}'_l, \bar{x}_h = \bar{x}'_m, \bar{z}_l = \bar{z}'_l, \bar{z}_h = \bar{z}'_m$;
6: **else**

7: $\quad \bar{x}_l = \bar{x}'_m, \bar{x}_h = \bar{x}'_h, \bar{z}_l = \bar{z}'_m, \bar{z}_h = \bar{z}'_h$
8: **end if**
9: **return** $\bar{x}_l, \bar{x}_h, \bar{z}_l, \bar{z}_h$.

**Determine** $\bar{x}_l, \bar{x}_h, \bar{z}_l, \bar{z}_h$ **given** $\bar{z}$.
1: $\bar{z}_l = \lfloor (2^h \cdot \bar{z})/2^h \rfloor, \bar{z}_h = \bar{z}_l + 2^{-h}$;
2: $\bar{x}_l = \overline{f^{-1}(\bar{z}_l)}, \bar{x}_h = \overline{f^{-1}(\bar{z}_h)}$;
3: **return** $\bar{x}_l, \bar{x}_h, \bar{z}_l, \bar{z}_h$.

**Interpolating uniform intervals in** $z$**.** This usually applies in the case that $f'$ is small (e.g. `sigmoid`). The uniform interval in co-domain $z$ is defined as $\bar{z}_l = \lfloor (2^h \cdot \bar{z})/2^h \rfloor, \bar{z}_h = \bar{z}_l + 2^{-h}$. The corresponding interval in the domain is $\bar{x}_l = \overline{f^{-1}(\bar{z}_l)}, \bar{x}_h = \overline{f^{-1}(\bar{z}_h)}$. For the inverse pass given $\bar{z}$, the interval can be obtained as above. For the forward pass, we first compute $z' = f(\bar{x})$, and then compute $\bar{z}'_m = \text{round}(2^h \cdot z')/2^h, \bar{z}'_l = \bar{z}'_m - 2^{-h}, \bar{z}'_h = \bar{z}'_m + 2^{-h}$, and $\bar{x}_{\{l,m,h\}} = \overline{f^{-1}(\bar{z}'_{\{l,m,h\}})}$. If $\bar{x} < \bar{x}_m$, we set the interval $\bar{z}_l = \bar{z}'_l, \bar{z}_h = \bar{z}'_m$; otherwise $\bar{z}_l = \bar{z}'_m, \bar{z}_h = \bar{z}'_h$. The method is summarized in Alg. 7.

Similarly as in Alg. 6, the correctness of Alg. 7 is ensured by $h \ll k$ and the limited numerical errors. The proof is very similar as to that of Alg. 6.

## C Extensions of iFlow

---

**Algorithm 8** Lossless Compression with Flows.

**Encode** $\mathbf{x}^\circ$.
1: Decode $\bar{\mathbf{v}}$ using $q(\bar{\mathbf{v}}|\mathbf{x}^\circ)\delta$;
2: $\bar{\mathbf{z}} \leftarrow \bar{f}(\bar{\mathbf{v}})$;
3: Encode $\bar{\mathbf{z}}$ using $p_Z(\bar{\mathbf{z}})\delta$;
4: Encode $\mathbf{x}^\circ$ using $P(\mathbf{x}^\circ|\bar{\mathbf{v}})$.

**Decode.**
1: Decode $\bar{\mathbf{z}}$ using $p_Z(\bar{\mathbf{z}})\delta$;
2: $\bar{\mathbf{v}} \leftarrow \bar{f}^{-1}(\bar{\mathbf{z}})$;
3: Decode $\mathbf{x}^\circ$ using $P(\mathbf{x}^\circ|\bar{\mathbf{v}})$;
4: Encode $\bar{\mathbf{v}}$ using $q(\bar{\mathbf{v}}|\mathbf{x}^\circ)\delta$;
5: **return** $\mathbf{x}^\circ$.

The extension of iFlow for lossless compression is summarized in Alg. 8. Note that for **Variational Dequantization Flow [19, 17] (Alg. 4)**, $\mathbf{v} = \mathbf{x}^\circ + \mathbf{u}(\mathbf{u} \in [0,1)^d)$, $q(\mathbf{v}|\mathbf{x}^\circ) = q(\mathbf{u}|\mathbf{x}^\circ)$, $P(\mathbf{x}^\circ|\mathbf{v}) = 1$. Thus the above coding procedure reduces to Alg. 4.

# D   Dynamic Uniform Entropy Coder in AC, ANS and UBCS

In this section we will demonstrate the effectiveness of UBCS compared with AC and ANS. Both AC and ANS use probability mass function (PMF) and cumulative distribution function (CDF) for encoding and inverse CDF for decoding. For ease of coding, the PMF and CDF are all mapped to integers in $[0, m)$. For uniform distribution $U(0, R)$ in which $P(s) = 1/R, s \in \{0, 1, ..., R-1\}$, the most simple way to compute PMF and CDF are

$$l_s = \text{PMF}(s) = \begin{cases} \lfloor m/R \rfloor, & s < R-1 \\ m - (R-1) \cdot \lfloor m/R \rfloor, & s = R-1 \end{cases},$$

$$b_s = \text{CDF}(s-1) = \lfloor m/R \rfloor \cdot s$$

$$(16)$$

Given $b \in [0, m)$, the output of inverse CDF $s = \text{CDF}^{-1}(b)$, should be exactly $b \in [b_s, b_s + l_s)$. The general way to determine $\text{CDF}^{-1}$ is binary search. But for uniform distribution, we can directly obtain the inverse CDF such that

$$s = \text{CDF}^{-1}(b) = \min(\lfloor b/l \rfloor, R-1), \quad l = \lfloor m/R \rfloor \quad (17)$$

We summarize uniform entropy coder AC, rANS and UBCS as follows:

For AC:

- **Initial state**: interval $[lo, hi)$;
- **Encoding**: get $m = hi - ho$, get $l_s, b_s$ with Eq. (16), update interval $[lo', hi')$ such that $lo' = lo + b_s, hi' = lo' + l_s$, get $c' \in [lo', hi')$ as the encoded bits;
- **Decoding**: for encoded bits $c' \in [lo, hi)$ and current interval $[lo, hi)$, get $m = hi - lo, b = c' - lo$, decode $s = \text{CDF}^{-1}(b)$ with Eq. (17), update interval $[lo', hi')$ such that $lo' = lo + b_s, hi' = lo' + l_s$;
- **Number of atom operations in encoding**: one division, one multiplication[3];
- **Number of atom operations in decoding**: two divisions, one multiplication. Binary search may involve if Eq. (17) is not used.

For rANS:

- **Initial state**: number $c$;
- **Encoding**: set $m = 2^K$, get $l_s, b_s$ with Eq. (16), update $c' = \lfloor c/l_s \rfloor \cdot m + (c \mod l_s) + b_s$;
- **Decoding**: set $m = 2^K$, for encoded bits $c'$, get $b = c' \mod m$, decode $s = \text{CDF}^{-1}(b)$ with Eq. (17), update $c = l_s \cdot \lfloor c'/m \rfloor + (c' \mod m) - b_s$;
- **Number of atom operations in encoding**: two divisions, two multiplications[4];
- **Number of atom operations in decoding**: two divisions, one multiplication. Binary search may involve if Eq. (17) is not used.

For UBCS:

- **Initial state**: number $c$;
- **Encoding**: update $c' = c \cdot R + s$;
- **Decoding**: for encoded bits $c'$, decode $s = c' \mod R$, update $c = \lfloor c'/R \rfloor$;
- **Number of atom operations in encoding**: one multiplication;

---

[3]We omit add/sub operations as they are negligible compared with multiplication/division.
[4]The multiplication/division/mod with $m = 2^K$ only involve bit operations. With the result of $\lfloor c/l_s \rfloor$, $c \mod l_s$ only involve one multiplication.

Table 4: More results on coding bandwidth (M symbol/s) of UBCS and rANS coder. We use the implementations in [17] for evaluating rANS.

|         | # threads | rANS             | UBCS               |
|---------|-----------|------------------|--------------------|
| Encoder | 1         | $5.1_{\pm 0.3}$  | $\mathbf{380}_{\pm 5}$     |
|         | 4         | $10.8_{\pm 1.9}$ | $\mathbf{709}_{\pm 56}$    |
|         | 8         | $15.9_{\pm 1.4}$ | $\mathbf{1297}_{\pm 137}$  |
|         | 16        | $21.6_{\pm 1.1}$ | $\mathbf{2075}_{\pm 353}$  |
| Decoder | 1         | $0.80_{\pm 0.02}$| $\mathbf{66.2}_{\pm 1.7}$  |
|         | 4         | $2.8_{\pm 0.1}$  | $\mathbf{248}_{\pm 8}$     |
|         | 8         | $5.5_{\pm 0.2}$  | $\mathbf{460}_{\pm 16}$    |
|         | 16        | $7.4_{\pm 0.5}$  | $\mathbf{552}_{\pm 50}$    |

- **Number of atom operations in decoding**: one division with remainder.

Overall, UBCS uses the least number of atom operations, which conveys that UBCS performs the best. Moreover, with PMF and CDF in Eq. (16), the optimal entropy coder cannot be guaranteed.

# E  More Experiments

In this section we will demonstrate our performance attributes across more benchmarking datasets. All experiments are conducted in PyTorch framework on one NVIDIA Tesla P100 GPU and Intel(R) Xeon(R) CPU E5-2690 @ 2.60GHz CPU. The code for LBB and the Flow++ model is directly taken from the original paper under MIT license.

## E.1  Coding Efficiency of UBCS

Table 5: Detailed results on the coding performance of iFlow and LBB on ImageNet32. We use a batch size of 64.

| flow arch. | compression technique | nll | bpd | aux. bits | encoding time (ms) | | decoding time (ms) | |
|------------|-----------------------|-----|-----|-----------|--------------------|--------------------|--------------------|--------------------|
|            |                       |     |     |           | inference          | coding             | inference          | coding             |
| Flow++     | LBB [17]              | 3.871 | 3.875 | 45.96 | $58.7_{\pm 0.1}$ | $176_{\pm 2.8}$ | $83.2_{\pm 0.4}$ | $172_{\pm 4.7}$ |
|            | **iFlow (Ours)**      |       | **3.873** | **34.40** |           | $\mathbf{66.6}_{\pm 0.3}$ |           | $\mathbf{95.3}_{\pm 0.3}$ |

Table 6: Detailed results on the coding performance of iFlow and LBB on ImageNet64. We use a batch size of 64.

| flow arch. | compression technique | nll | bpd | aux. bits | encoding time (ms) | | decoding time (ms) | |
|------------|-----------------------|-----|-----|-----------|--------------------|--------------------|--------------------|--------------------|
|            |                       |     |     |           | inference          | coding             | inference          | coding             |
| Flow++     | LBB [17]              | 3.701 | **3.703** | 38.00 | $24.4_{\pm 0.0}$ | $284_{\pm 2.5}$ | $35.7_{\pm 0.1}$ | $281_{\pm 2.2}$ |
|            | **iFlow (Ours)**      |       | **3.703** | **34.42** |           | $\mathbf{45.0}_{\pm 1.7}$ |           | $\mathbf{57.0}_{\pm 1.4}$ |

The detailed experiments is shown in Table 4.

## E.2  Compression Performance

Detailed experimental results on ImageNet32 and ImageNet64 datasets are displayed in Tables 5 and 6. Note that we further report the decoding time, which we observe is close to the model inference time.

## E.3  Hyper-parameters

As discussed in Sec. 2.4, the codelength will be affected by the choices of $h, k$ and $S$. As $S$ is set to a large value – and will minimally affect the codelength resulting from MST – we mainly discuss $h$ and $k$. Tables 7, 8 and 9 illustrate the codelength and auxiliary bits (in bpd) for differing choices

of $h$ and $k$. It is clear that, for large $k$, the codelength decreases with a larger $h$. This is expected as larger $h$ corresponds to a greater numerical precision of our linear interpolation. On the other hand, for a fixed $h$, the codelength becomes larger with a smaller $k$, as a smaller $k$ corresponds to a greater quantization error. A smaller $k$ may even lead to the failure of iFlow entirely – especially if $k$ is close to $h$, which would result in the potential of a zero-valued $R$ in Eq. (4) for sufficiently small $|f'(\bar{x})|$. On the other hand, the auxiliary bits are principally affected by $k$ and not $h$. Therefore we note that a smaller $k$ is preferred. To conclude, we can nonetheless achieve a near-optimal codelength with a considered choice of hyper-parameters. Thus we set $k = 28$ and $h = 12$ for the experiments.

Table 7: Codelengths in terms of bpd and auxiliary length on different $h$ and $k$ on the CIFAR10 dataset. N/A denotes the failure of the compression procedure. The theoretical bpd (nll) is **3.116**.

| | | $h$ | | | | |
| | | 6 | 8 | 10 | 12 | 14 |
|---|---|---|---|---|---|---|
| | | bpd | | | | |
| | 18 | $3.229_{\pm 0.000}$ | N/A | N/A | N/A | N/A |
| | 20 | $3.225_{\pm 0.000}$ | $3.152_{\pm 0.000}$ | N/A | N/A | N/A |
| | 22 | $3.224_{\pm 0.000}$ | $3.147_{\pm 0.000}$ | $3.130_{\pm 0.000}$ | N/A | N/A |
| $k$ | 24 | $3.224_{\pm 0.000}$ | $3.146_{\pm 0.000}$ | $3.126_{\pm 0.000}$ | $3.124_{\pm 0.000}$ | N/A |
| | 26 | $3.224_{\pm 0.000}$ | $3.146_{\pm 0.000}$ | $3.125_{\pm 0.000}$ | $3.119_{\pm 0.000}$ | $3.122_{\pm 0.000}$ |
| | 28 | $3.224_{\pm 0.000}$ | $3.146_{\pm 0.000}$ | $3.124_{\pm 0.000}$ | $3.118_{\pm 0.000}$ | $3.118_{\pm 0.000}$ |
| | 30 | $3.224_{\pm 0.000}$ | $3.146_{\pm 0.000}$ | $3.124_{\pm 0.000}$ | $3.118_{\pm 0.000}$ | $3.116_{\pm 0.000}$ |
| | 32 | $3.224_{\pm 0.000}$ | $3.146_{\pm 0.000}$ | $3.124_{\pm 0.000}$ | $3.118_{\pm 0.000}$ | $3.116_{\pm 0.000}$ |
| | | auxiliary length | | | | |
| | 18 | $24.25_{\pm 0.01}$ | N/A | N/A | N/A | N/A |
| | 20 | $26.25_{\pm 0.01}$ | $26.27_{\pm 0.01}$ | N/A | N/A | N/A |
| | 22 | $28.26_{\pm 0.01}$ | $28.27_{\pm 0.01}$ | $28.27_{\pm 0.01}$ | N/A | N/A |
| $k$ | 24 | $30.25_{\pm 0.01}$ | $30.27_{\pm 0.01}$ | $30.27_{\pm 0.01}$ | $30.27_{\pm 0.01}$ | N/A |
| | 26 | $32.25_{\pm 0.01}$ | $32.27_{\pm 0.01}$ | $32.27_{\pm 0.01}$ | $32.27_{\pm 0.01}$ | $32.27_{\pm 0.01}$ |
| | 28 | $34.25_{\pm 0.01}$ | $34.27_{\pm 0.01}$ | $34.27_{\pm 0.01}$ | $34.27_{\pm 0.01}$ | $34.27_{\pm 0.01}$ |
| | 30 | $36.25_{\pm 0.01}$ | $36.26_{\pm 0.01}$ | $36.27_{\pm 0.01}$ | $36.27_{\pm 0.01}$ | $36.27_{\pm 0.01}$ |
| | 32 | $38.25_{\pm 0.01}$ | $38.26_{\pm 0.01}$ | $38.27_{\pm 0.01}$ | $38.27_{\pm 0.01}$ | $38.27_{\pm 0.01}$ |

Table 8: Codelengths in terms of bpd and auxiliary length on different $h$ and $k$ on a **SUBSET** of ImageNet32 dataset. N/A denotes the failure of the compression procedure. The theoretical bpd (nll) is **3.883**.

| | | $h$ | | | | |
| | | 6 | 8 | 10 | 12 | 14 |
|---|---|---|---|---|---|---|
| | | bpd | | | | |
| | 18 | $3.994_{\pm 0.000}$ | $3.953_{\pm 0.000}$ | N/A | N/A | N/A |
| | 20 | $3.985_{\pm 0.000}$ | $3.919_{\pm 0.000}$ | N/A | N/A | N/A |
| | 22 | $3.983_{\pm 0.000}$ | $3.910_{\pm 0.000}$ | $3.900_{\pm 0.000}$ | N/A | N/A |
| $k$ | 24 | $3.983_{\pm 0.000}$ | $3.908_{\pm 0.000}$ | $3.892_{\pm 0.000}$ | $3.896_{\pm 0.000}$ | $3.928_{\pm 0.000}$ |
| | 26 | $3.983_{\pm 0.000}$ | $3.908_{\pm 0.000}$ | $3.890_{\pm 0.000}$ | $3.887_{\pm 0.000}$ | $3.894_{\pm 0.000}$ |
| | 28 | $3.983_{\pm 0.000}$ | $3.908_{\pm 0.000}$ | $3.889_{\pm 0.000}$ | $3.885_{\pm 0.000}$ | $3.886_{\pm 0.000}$ |
| | 30 | $3.982_{\pm 0.000}$ | $3.908_{\pm 0.000}$ | $3.889_{\pm 0.000}$ | $3.885_{\pm 0.000}$ | $3.884_{\pm 0.000}$ |
| | 32 | $3.983_{\pm 0.000}$ | $3.908_{\pm 0.000}$ | $3.889_{\pm 0.000}$ | $3.885_{\pm 0.000}$ | $3.884_{\pm 0.000}$ |
| | | auxiliary length | | | | |
| | 18 | $24.37_{\pm 0.01}$ | $24.37_{\pm 0.01}$ | N/A | N/A | N/A |
| | 20 | $26.38_{\pm 0.01}$ | $26.39_{\pm 0.01}$ | N/A | N/A | N/A |
| | 22 | $28.38_{\pm 0.01}$ | $28.39_{\pm 0.01}$ | $28.39_{\pm 0.01}$ | N/A | N/A |
| $k$ | 24 | $30.38_{\pm 0.01}$ | $30.39_{\pm 0.01}$ | $30.40_{\pm 0.01}$ | $30.39_{\pm 0.01}$ | $30.38_{\pm 0.01}$ |
| | 26 | $32.38_{\pm 0.01}$ | $32.39_{\pm 0.01}$ | $32.40_{\pm 0.01}$ | $32.40_{\pm 0.01}$ | $32.39_{\pm 0.01}$ |
| | 28 | $34.38_{\pm 0.01}$ | $34.39_{\pm 0.01}$ | $34.40_{\pm 0.01}$ | $34.40_{\pm 0.01}$ | $34.40_{\pm 0.01}$ |
| | 30 | $36.38_{\pm 0.01}$ | $36.39_{\pm 0.01}$ | $36.39_{\pm 0.01}$ | $36.40_{\pm 0.01}$ | $36.40_{\pm 0.01}$ |
| | 32 | $38.38_{\pm 0.01}$ | $38.39_{\pm 0.01}$ | $38.39_{\pm 0.01}$ | $38.39_{\pm 0.01}$ | $38.39_{\pm 0.01}$ |

Table 9: Codelengths in terms of bpd and auxiliary length on different $h$ and $k$ on a **SUBSET** of ImageNet64 dataset. N/A denotes the failure of the compression procedure. The theoretical bpd (nll) is **3.718**.

|   |   | $h$ | | | | |
|---|---|---|---|---|---|---|
|   |   | 6 | 8 | 10 | 12 | 14 |
|   |   | | | bpd | | |
| $k$ | 18 | $3.829_{\pm0.000}$ | $3.779_{\pm0.000}$ | $3.867_{\pm0.000}$ | N/A | N/A |
|   | 20 | $3.823_{\pm0.000}$ | $3.753_{\pm0.000}$ | $3.760_{\pm0.000}$ | $3.863_{\pm0.000}$ | N/A |
|   | 22 | $3.821_{\pm0.000}$ | $3.746_{\pm0.000}$ | $3.733_{\pm0.000}$ | $3.755_{\pm0.000}$ | $3.861_{\pm0.000}$ |
|   | 24 | $3.821_{\pm0.000}$ | $3.744_{\pm0.000}$ | $3.727_{\pm0.000}$ | $3.729_{\pm0.000}$ | $3.754_{\pm0.000}$ |
|   | 26 | $3.821_{\pm0.000}$ | $3.744_{\pm0.000}$ | $3.725_{\pm0.000}$ | $3.722_{\pm0.000}$ | $3.727_{\pm0.000}$ |
|   | 28 | $3.821_{\pm0.000}$ | $3.744_{\pm0.000}$ | $3.725_{\pm0.000}$ | $3.720_{\pm0.000}$ | $3.721_{\pm0.000}$ |
|   | 30 | $3.821_{\pm0.000}$ | $3.744_{\pm0.000}$ | $3.725_{\pm0.000}$ | $3.720_{\pm0.000}$ | $3.719_{\pm0.000}$ |
|   | 32 | $3.821_{\pm0.000}$ | $3.744_{\pm0.000}$ | $3.725_{\pm0.000}$ | $3.720_{\pm0.000}$ | $3.719_{\pm0.000}$ |
|   |   | | | auxiliary length | | |
| $k$ | 18 | $24.40_{\pm0.01}$ | $24.41_{\pm0.01}$ | $24.39_{\pm0.01}$ | N/A | N/A |
|   | 20 | $26.40_{\pm0.01}$ | $26.41_{\pm0.01}$ | $26.41_{\pm0.01}$ | $26.39_{\pm0.01}$ | N/A |
|   | 22 | $28.40_{\pm0.01}$ | $28.42_{\pm0.01}$ | $28.42_{\pm0.01}$ | $28.41_{\pm0.01}$ | $28.39_{\pm0.01}$ |
|   | 24 | $30.40_{\pm0.01}$ | $30.42_{\pm0.01}$ | $30.42_{\pm0.01}$ | $30.42_{\pm0.01}$ | $30.41_{\pm0.01}$ |
|   | 26 | $32.40_{\pm0.01}$ | $32.42_{\pm0.01}$ | $32.42_{\pm0.01}$ | $32.42_{\pm0.01}$ | $32.42_{\pm0.01}$ |
|   | 28 | $34.40_{\pm0.01}$ | $34.42_{\pm0.01}$ | $34.42_{\pm0.01}$ | $34.42_{\pm0.01}$ | $34.42_{\pm0.01}$ |
|   | 30 | $36.40_{\pm0.01}$ | $36.42_{\pm0.01}$ | $36.42_{\pm0.01}$ | $36.42_{\pm0.01}$ | $36.42_{\pm0.01}$ |
|   | 32 | $38.40_{\pm0.01}$ | $38.42_{\pm0.01}$ | $38.42_{\pm0.01}$ | $38.42_{\pm0.01}$ | $38.42_{\pm0.01}$ |