# OpenReview forum: "iFlow: Numerically Invertible Flows for Efficient Lossless Compression via a Uniform Coder"
_NeurIPS.cc/2021/Conference — NeurIPS 2021 Spotlight_

### Official Review · Reviewer_BnZR · 2021-06-29

**Rating:** 6
**Confidence:** 2

**Summary:**

The paper introduces a new normalizing flow method for lossless compression. The key innovation is a new family of techniques to convert a continuous flow into a discrete bijector, which can transform binary codes without information loss. The authors compare compression performance and encoding speed with a large family of state-of-the-art neural compression methods.

**Limitations And Societal Impact:**

Limitations and social impact are  discussed, albeit briefly.

**Main Review:**

The authors introduce a method to convert normalizing flows into discrete bijectors that can be used for lossless compression. This allows for the use of a trained continuous flow for lossless compression without lossy quantization of the encoded latent code. Specifically, the authors introduce the "modular scale transform", which approximate element-wise scalar multiplications with integer multiplications and divisions plus a remainder that is encoded separately and follows a uniform distribution.

I premise that my expertise in the area is limited. So said, the work seems to be solid and the theoretical results straightforward. The main innovation is the modular scale transform, which is a very simple trick. This simplicity however is not necessarily a problem as far as the innovation is original. In the following, I will list the main strengths and weaknesses of the paper.

Strengths:
- The main innovation is simple and elegant and has a wide applicability. The modular scale transform is used to approximate arbitrary monotonic transformation through locally-linear approximations and then complex normalizing flow layers using a similar approach.
- The experiments are performed on a rather large number of datasets and comparison are made with a large number of relevant baselines.

Weakness:
- The main contribution is a very simple trick. Given that the field is young, I can image that it is original. However, it is possible that a very similar method is already in wide use in a related literature. Given my scarce familiarity with the compression literature, I trust the judgment of the other reviewers on this matter.
- The experiments are difficult to interpret as the authors do not provide error bars on their result. I suggest to use a bootstrap technique in order to quantify the statistical variability of the performance.




**Time Spent Reviewing:**

2

---

> ### Author Response · Authors · 2021-08-10
> **Response to Reviewer-BnZR**
>
> Thank you for your comments. We address some of your concerns as follows:
> 1.	(The originality of the iFlow) MST is a simple but novel algorithm. It forms the basis of iFlow by extending to non-linear flow transformations and practical flow layers. The most relevant work is MAT in iVPF, and the differences between MST and MAT are discussed in Reviewer-EPj6.
> 2.	(About the experiments) We show the error bars in the supplementary material, including the coding time and the compression length. This will be included in the main text in the final version.

---

> > ### Comment · Reviewer_BnZR · 2021-08-12
> > **Reply**
> >
> > Thank you for the response. I am satisfied by the response and I vote for accepting. I would suggest to include the error bars in the main document next time as they are essential in order to evaluate the results.

---

### Official Review · Reviewer_Z7Bx · 2021-07-16

**Rating:** 9
**Confidence:** 3

**Summary:**

This paper proposes a lossless image compression method based on normalizing flows. In order for this class of flows to be useful for compression, they need to be working in the quantized domain, and (of course) have to be invertible. The authors present an extension to a much more general class of functions that can be used in the composition formulation of the flow, therefore being able to achieve higher rates of compression than what was possible before. Additional contributions include using a variant of bits-back coding to achieve higher compression rates, and a much faster entropy coding method that vastly outperforms rANS.

**Ethical Concerns:**

No issues.

**Limitations And Societal Impact:**

This looks ok.

**Main Review:**

This paper could be the start of a many more papers using flows for compression. Thus far there are very few papers published in this area that use normalizing flows due to the very challenging nature of the problem. The key blocking factor that this paper addresses is how to use more general composable functions than what was possible before (which were very weak functions). This improvement might make this paper a turning point in utilization of this class of methods.

While I did try to do my best to understand the entirety of the paper, I did not follow the proofs closely as I am not super-familiar with the vast amount of background knowledge needed to understand methods ranging from flows, which I am familiar with, to entropy coding and bits-back coding (which I am much less familiar with). I budgeted about 3 hours for this paper, and it was really not enough to get fully up to speed with everything needed here.

With that said, the main body of the paper was very clearly written, and I wish more papers were as well written as this one. The supplementary materials contain a vast amount of proofs, and I think this is very welcomed given the vast quantity of information presented in this work.

The results are state o the art, and the only criticism I can find is that the decoding/encoding time is only listed for toy problems. With that said, the number of baselines, including state of the art classical codecs is very impressive and very welcomed.

**Time Spent Reviewing:**

3

---

> ### Author Response · Authors · 2021-08-10
> **Response to Reviewer-Z7Bx**
>
> Thank you for your comments. We address some of your concerns as follows:
> 1.	Bits-back coding is a certain entropy coding algorithm used for variational models, in which “bits-back” decoding is performed before encoding the data to achieve lower compression length. The expected codelength is ELBO. More details of bits-back can be found in [1].
> 2.	With the flow model, for the encoding process, we convert $x$ to a latent variable $z$. We then code $z$ using some prior distribution $p(z)$. For decoding, $z$ is firstly decoded using $p(z)$, where we then recover $x$ with the inverse flow. As both $x$ and $z$ are discrete, we must define an invertible mapping between discrete $x$ and $z$, which is in intractable for the most popular classes of flows. iFlow solve this issue with application to lossless compression. This is our main contribution.
> 3.	As we crop the data to patches for compression, the coding time is proportional to the number of pixels. See Reviewer-Poho-Q4 for more details.
>
> [1] Townsend J, Bird T, Barber D. Practical lossless compression with latent variables using bits back coding. arXiv preprint arXiv:1901.04866. 2019 Jan 15.

---

> > ### Comment · Reviewer_Z7Bx · 2021-08-17
> > **Reply**
> >
> > Thanks for the reply/reference. I am still quite excited by this work!

---

### Official Review · Reviewer_EPj6 · 2021-07-16

**Rating:** 6
**Confidence:** 3

**Summary:**

This paper proposes iFlow - a method to improve lossless compression using flow-based generative models. Specifically, the paper proposes two novel methods, the Modular Scale Transform (MST) and Uniform Base Conversion Systems (UBCS). MST is a method of obtaining a numerically invertible scale transform, to be used as a building block for flow layers. UBCS is a codec for uniformly distributed symbols, designed for speed.

The authors demonstrate that the MST results in superior bitrates when compared to similar techniques, and that UBCS is faster for coding uniform symbols than competing entropy codecs.

**Ethical Concerns:**

I don't see any ethical concerns with this work.

**Limitations And Societal Impact:**

They have addressed the limitations. I don't see any negative societal impact.

**Main Review:**

The results in this work appear to be quite interesting. There does appear to be an advance in coding rate and compression speed for lossless compression using flow models (or even any probabilistic generative model, since flow models are currently the SOTA). The realisation that the majority of the coding can be done via uniformly distributed symbols is neat, and then showing that a simple codec (UBCS) can code these close to the Shannon entropy much faster than a modern codec like rANS is I think an important realisation. This could potentially have important ramifications for the field of learned lossless compression as previous works have not utilised this. For example [1] uses a uniform codec for the latents since a Gaussian is discretised into buckets of uniform probability. They use rANS for the coding though, which appears not to be optimal in terms of speed.

This work is in general well-written and precise in its statements. Results are given clearly and theorems and motivations are well laid out.

The concern I have with this work is that it is extremely close in content to [2]. [2] is a recent paper but it is cited and clearly built upon in this work. One of the major contributions of this work is the MST, which the authors admit is inspired by MAT (just changing affine to scalar) from [2]. They look very similar to me, and although the schemes are clearly slightly different I can't quite work out why MST is superior, and it doesn't seem to be stated in this paper. Could the authors comment and make clear exactly why MST is an iteration over MAT?

Much like in [2], after describing the MST, time is then taken to show that various flow compression schemes can be implemented using these numerically invertible flows. It doesn't appear that much has changed between these derivations.

UBCS is still a novelty, but it is in someways quite a simple revelation - we can code uniform symbols fast. The codec itself is straightforward. This is not a bad thing, but with the rest of the paper being very closely related to another work, it does detract from the overall novelty somewhat.

If the authors addressed exactly the motivation for MST over MAT then I consider upgrading my score.

Edit: after response I am upgrading my score as per the above.

[1] Practical Lossless Compression with Latent Variables using Bits Back Coding
James Townsend, Tom Bird, David Barber
ICLR 2019

[2] iVPF: Numerical Invertible Volume Preserving Flow for Efficient Lossless Compression
Shifeng Zhang, Chen Zhang, Ning Kang, Zhenguo Li
CVPR 2021


**Time Spent Reviewing:**

3

---

> ### Author Response · Authors · 2021-08-10
> **Response to Reviewer-EPj6: The Contribution of MST**
>
> Thank you for your comments. The reviewer mainly raises concerns of MST’s novelty as compared with MAT in iVPF. Although both MAT and MST involve integer multiplication and division with remainder, we believe MST is able to work in more general cases than MAT, along with demonstrated greater computational efficiency. We detail our perspective below:
> 1.	MAT is only able to deal with volume-preserving affine transformations $z=s \cdot x (\prod s = 1)$. To be specific, it must hold that $m_0=m_{d_b}$ so that the remainder can spread between coupling layers. While MST is more flexible, and achieves numerical invertibility for any affine transformation $z=s \cdot x$ without additional constraints.
> 2.	MAT is performed sequentially along dimensions, as the remainder in the last dimension should be fed as input to the next dimension. MST is much simpler, and each dimension of affine transform can be computed in parallel. Thus MST is more efficient than MAT. Experiments (in Table 2) show that by replacing MAT with MST in iVPF, we are able to achieve an improved coding efficiency.
> 3.	Another key contribution of iFlow is that we are able to achieve numerically invertible non-linear transformations based on MST (Sec. 2.2.2). With non-linear transformations, we can derive a lossless compression algorithm with almost any type of flow. In contrast, MAT cannot extend to the non-linear case, in the sense that the volume-preserving affine transformation is almost not hold when linear interpolating non-linear functions. Overall, MAT is restricted to the class of volume-preserving flows, limiting the expressive power and resulting compression performance.

---

> > ### Comment · Reviewer_EPj6 · 2021-08-20
> > **Reply to respose**
> >
> > Thanks for your reply. I think I follow your points, and the improvement of MST over MAT does seem to be meaningful. I will therefore raise my score to 6.
> >
> > I would appreciate if the you added some version of this explanation to the manuscript to make this clear to the reader, since I think these points are non-obvious.

---

### Official Review · Reviewer_hYZr · 2021-07-17

**Rating:** 8
**Confidence:** 3

**Summary:**

The two key contributions in this work are:

1) A construction method for low-error discretized approximations to a wide class of continuous normalizing flow layers
2) An entropy coding method the discrete uniform distribution that the authors call Uniform Base Conversion Systems (UBCS) that is simple to implement and is very fast.

The authors combine these two contributions in their experiments and achieve state-of-the-art compression ratios on the standard lossless image compression benchmark datasets (ImageNet, Cifar10, CLIC, DIV2K), while also achieving large gains in terms of compression speed compared to other methods.


**Limitations And Societal Impact:**

They have.

**Main Review:**

# Strengths
There are several novel ideas presented in the paper. The modular scale transform (MST) is simple, easy to implement and the authors show using some results from [1] that it has desirable properties. Then, the authors show how MST can be used as the fundamental building block for discrete approximations of general elementwise flows which then leads to autoregressive and 1 x 1 convolutional flows, the correctness and efficiency of which is all guaranteed by the correctness and efficiency of MST.

The authors adopt some ideas from [2] to remove the overhead introduced by the quantization mesh and call it bits-back dequantization.
The authors introduce Uniform Base Conversion Systems (UBCS). While a very simple scheme, it is very fast and just the right entropy coding method to use in conjunction with the discrete flow approximation proposed by the authors.

The experimental method in this work is standard for learned lossless image compression, and the authors achieve very good results on all considered datasets. Concretely, they match or slightly improve the current state-of-the-art compression performance on all datasets, while achieving a large improvement in terms of run-time compared to other methods.

The paper is well-written with a logical and easy-to-follow layout. Nice submission overall.

[1] Shifeng Zhang, Chen Zhang, Ning Kang, and Zhenguo Li. iVPF: Numerical invertible volume preserving flow for efficient lossless compression. arXiv 2021

[2] Jonathan Ho, Evan Lohn, and Pieter Abbeel. Compression with flows via local bits-back coding. NeurIPS 2019

# Weaknesses
There are no major weaknesses that I could spot. There are some minor issues, concretely:
- In the abstract it says "We first develop UBCS, ...", but in fact it is developed after the introduction of iFlow.
- On several occasions (e.g. in Algorithms 1 and 4) the authors mention "decoding" an object from a distribution.
  I believe that the authors here refer to the fact that entropy coding methods can be viewed as "invertible samplers",
  where "decoding" can be identified with sampling the probability distribution at hand using the currently available bit stream.
  I don't think this is clarified anywhere in the main text, and I also don't think that this is common terminology in the wider community. Therefore I would suggest that the authors either clarify it or change their wording.
- Footnote 2: I think the language used is imprecise, I think the authors mean that the intervals must _partition_ the domain of $f$.
- Could the authors clarify what the relevance of Eq (4) is? Is it to show that $R$ in the MST implementation of Eq (3) will have the appropriate magnitude so that Prop (2) holds?
- Line 59: "Jacabian matrix"
- Line 115: subscript of $z$ in $\bar{z}_l = f(\bar{x}_h)$ should be $h$.


**Time Spent Reviewing:**

4-5

---

> ### Author Response · Authors · 2021-08-10
> **Response to Reviewer-hYZr**
>
> Thank you for your comment and we will make relevant amendments in the final version. We will address the issues as follows:
> 1.	(About the contribution of the paper) We will reformulate the contribution in the final version. In fact, iFlow is our main contribution, and UBCS is the key novelty in the iFlow model.
> 2.	(The terminology of entropy coding) For any entropy coder method, “encoding” means to encode $x$ to the bit-stream using the distribution $p(x)$; and “decoding” means to decode $x$ from the bit-stream using the distribution $p(x)$. In the paper, it is better to say “Decode/Encode $x$ using $p(x)$ from the bit-stream” in Alg. 1/4, which is also used in LBB [1]. We will make this clear in the final version.
> 3.	(About the Footnote 2) Thank you for your comment, and we will adopt your advice in the final version.
> 4.	(Correctness of Eq. (4)) $R$ is set as Eq. (4) so that $R/S$ approximates $ (z_h – z_l)  / (x_h – x_l) $ (discussed in Supplementary A.4) and $\bar{f}_{inp} (\bar{x}_h – 2^{-k}) < \bar{z}_h$ (discussed in L117-119).
> 5.	(Other comments) Typos in L59 and L115 will be corrected in the final version.
>
> [1] Ho J, Lohn E, Abbeel P. Compression with flows via local bits-back coding. arXiv preprint arXiv:1905.08500. 2019 May 21.

---

> > ### Comment · Reviewer_hYZr · 2021-08-18
> > **Response**
> >
> > Thank you for the authors' response. After reading the other reviews and the authors' rebuttal, I still firmly believe that the work is a very good contribution to the learned compression literature, and hence maintain my score.

---

### Official Review · Reviewer_PoNo · 2021-07-28

**Rating:** 8
**Confidence:** 3

**Summary:**

The paper performs lossless image compression with invertible flows. The authors propose a way to convert flows to be numerically invertible, and a coder for uniform distributions.

**Limitations And Societal Impact:**

The paper addresses this.

**Main Review:**

I'm reviewing this paper from a compression perspective. I know the invertible flow literature on a high level only. With that in mind, the idea that is used to convert flows makes sense, and is novel as far as I know. I like the formulation of the paper, it is concise but understandable.

My main concern is that I do not understand the point and/or novelty of the proposed "Uniform Base Conversion System". I have two concerns:
- [Update: addressed by rebuttal] Are we using UBCS also to encoding the final $z$? It seems suboptimal to assume $z\sim U$. Could you elaborate on the distribution used for z? If it's not uniform, what coder do you use?
- [Update: addressed by rebuttal]  The goal is to compress symbols that are uniformly distributed according to $U(0, R)$. To me it seems many of the existing algorithms should be fast if adapted to this assumption. In particular, if there exists an $r$ such that $R=2^r$, you can write a simple algorithm in C that does not use any multiplication, just bitshifts and additions, that should be blazingly fast. For other $R$, I'm sure that the tabled variant of ANS, tANS, should also allow for very fast encoding and decodings, as it also does not use any multiplications -- in contrast to the presented algorithm. It seems unfair to compare to a particular implementation of rANS. Overall, this is maybe a nit-picky point, as the proposed variant is faster than the compared rANS, but it just seems that with a bit more tinkering, an even much faster variant could have been found. In light of what exists, claims of novelty should probably be toned down. Also, L293, "... binary search protocol required at decode time ..." seems unfair, as AC can trivially be adapted to use a lookup table for fixed uniform distributions.

Given the comments on improved/practical speed in the Introduction, I would have liked to see a table with runtimes on full-resolution datasets such as CLIC.pro.

Minor comments:
- L82: why is a triple equality used.
- L85: Footnote could be inlined.
- L185 could use a restatement of $\delta$ and $-\log \delta = kd$

**Time Spent Reviewing:**

2

---

> ### Author Response · Authors · 2021-08-10
> **Response to Reviewer-PoNo: Dynamic Entropy Coder in AI Lossless Compression and UBCS**
>
> Thank you for your comments. We further your concerns as follows, especially with respect to the UBCS algorithm:
> 1.	(Coding the final $z$) Here we use the rANS algorithm to encode the final $z$, as the prior of $z$ is either Gaussian (as in Flow++) or a mixture of Gaussians (as in iVPF). Given that we only perform rANS once, the latency is negligible.
> 2.	(The advantage of UBCS) It should be emphasized that AI compression (including iFlow) requires *dynamic* entropy coder, where the underlying distributions are modified for each coding process. In this work, $R$ (denoted in $U(0, R)$) varies for different instances of Alg. 1, in which the value of $R$ is determined by the input data and the model. Although tANS or static versions of AC/rANS are all much faster, they are not working well under following aspects: (1) $R=2^r$ mostly does not hold; (2) tANS likely struggles with a varied $U(0, R)$, as a number of tables should be established for each instance, resulting in potentially large memory costs; (3) in AC, numerous lookup tables may be built for each $R$; (4) whilst we avoid a binary search when coding a uniform distribution with rANS and AC (as the inverse CDF can be exactly computed), determining the inverse CDF involves at least 2 multiplications and divisions, which is more complex than the proposed UBCS. Overall, UBCS is most suitable for use as a *dynamic* uniform entropy coder, on which iFlow relies. We will more clearly emphasize the importance of *dynamic* entropy coder and its application to iFlow.
> 3.	(Coding time in high resolution images) We will show the runtime results associated with the compression of full-resolution images in the final version. In fact, as we crop the full-resolution images to patches for compression, the coding time is proportional to the number of pixels. For our ImageNet64 model, the encoding time is 45ms per 64x64 patch (see Table 6 in the Supplementary), thus encoding a 2048x1536 image (typical size of CLIC) uses approximately 34s. Note that, in contrast, LBB would require 218s.
> 4.	(Other comments) In L82, we emphasize that the recovered data $\bar{f}^{-1}(\bar{z})$ must be the same as the input data $x$. In L85 and L185, we agree with the reviewer’s suggestions and will amend for the final version.

---

> > ### Comment · Reviewer_PoNo · 2021-08-12
> > **Response**
> >
> > Thanks for your reply.
> >
> > Re. point 1: Thanks, this clarifies things.
> >
> > Re. point 2: It was unclear to me that R would consistently change, but it makes sense now. I think highlighting this would make sense. I'm still unsure on the exact computational complexity of UBCS and an adapted AC. My limited understanding gives:
> >
> > UBC:
> > - encode: for each R, and for each symbol: multiplications/divisions in L1 and L4 in Alg. 5 (assuming $2^K$ is a constant).
> > - decode: multiplications/divisions in L1, L2, L5
> >
> > AC:
> > - encode: for each R, and for each symbol: calculate `lower = sym * width ; upper = lower + width`, and the rest is bitshifts.
> > - decoder: `sym = lower / width` plus some bitshifts.
> >
> > I might be missing something, but I think the two algorithms should be in the same order in terms of complexity, and I believe this should be mentioned in the paper, and compared to with an actual benchmark. As it stands, the comparison to rANS in Table 1 seems misleading.
> >
> > I would also be satisfied if claims about novelty for UBCS are toned down and/or the option of using AC or similar is emphasized.
> >
> > Re. point 3: These numbers seem rather slow. I'm looking at L3C [29] another neural lossless compression paper, encoding a 512x512 image took 242ms, which seems to be 2880ms for you. L3C used a fully adaptive range coder, but is obviously worse in terms of bpd.

---

> > > ### Author Response · Authors · 2021-08-16
> > > **Response**
> > >
> > > Thank you for your further comment.
> > >
> > > Q2
> > >
> > > The main reasons why AC is not considered are as follows:
> > > 1. Coding with iFlow must perform in first-in-last-out as bits-back coding is involved (more details can be referred to in BB-ANS [1]). AC is coded in first-in-first-out manner, which is impractical in iFlow.
> > > 2. AC suffers from a waste of bits. The codelength is $\log (upper - lower) – \log \lfloor (upper – lower) / R \rfloor > \log R$ (`width` is computed as below, regardless of bitshifts). While the codelength of UBCS is $\log R$ (regardless of bitshifts). Thus uniform coding with UBCS will achieve smaller codelength.
> > >
> > > Besides these two points, UBCS is still superior to AC.
> > >
> > > For UBCS
> > >
> > > Encoding: only one multiplication is involved (in L1). As $2^K$ is constant, the division and mode operations with $2^K$ (L3-4) only involve bitwise operations (right shift in division, AND in mode), which is negligible compared with multiplications.
> > >
> > > Decoding: only one division with remainder is involved (in L4-5). The multiplications of $2^M, 2^K$ can be performed with bitwise operation (left shift), which is negligible compared with divisions.
> > >
> > > For AC
> > >
> > > Encoding: for each `R`, the `width` must be computed with one division operation such that `width = (upper – lower) / R` (`/` denotes rounding-down division). Then update `lower, upper` such that `lower = lower + sym * width, upper = lower + width`. Overall, one division and one multiplication are involved.
> > >
> > > Decoding: the width must be firstly computed with `width = (upper – lower) / R`, then get `sym` with `sym = (code - lower) / width` (`code` denotes the final encoded number) and update `lower = lower + sym * width, upper = lower + width`. Overall, two divisions and one multiplication are involved.
> > >
> > > Compared with UBCS, AC uses one more division operation in encoding and one more division/multiplication operations in decoding. Thus it is believed that UBCS performs the best.
> > >
> > > Q3
> > >
> > > Our paper aims at efficient AI lossless compression based on flow model, and treat flow-based codec (IDF, IDF++, LBB, iVPF) as the baselines. Experiments show the coding efficiency over the baselines.
> > >
> > > We adopt flow model for efficient lossless compression, in the sense that flow has much better potential than L3C in compression ratio. The flow model used is more complex than L3C, affecting the bandwidth. We can use smaller flow model for speed up without much drop in compression ratio (like binary flow model [2]).
> > >
> > > ---
> > >
> > > [1] Townsend J, Bird T, Barber D. Practical lossless compression with latent variables using bits back coding. arXiv preprint arXiv:1901.04866. 2019 Jan 15.
> > >
> > > [2] Bird T, Kingma FH, Barber D. Reducing the Computational Cost of Deep Generative Models with Binary Neural Networks. arXiv preprint arXiv:2010.13476. 2020 Oct 26.

---

> > > > ### Comment · Reviewer_PoNo · 2021-08-16
> > > > **Response**
> > > >
> > > > Thanks for the detailed reply. I am mostly satisfied with the answer, the only remaining gripe I have is with Table 1. I think a note on number of operations would be better, or to add an "optimized AC" column in the table. Either way, I'm changing my vote to accepting.

---

> > > > > ### Author Response · Authors · 2021-08-17
> > > > > **Response**
> > > > >
> > > > > Thank you for your comments and suggestions. We will refine our paper in the final version, including discussing the importance of *dynamic* entropy coder, more discussions on encoding time and advantage of UBCS, etc.

---

### Decision · Program_Chairs · 2021-09-27

**Decision:**

Accept (Spotlight)

**Comment:**

The main concerns by the reviewers were adequately addressed during the rebuttal and subsequent discussion between the authors and reviewers. The reviewers unanimously vote for accepting this paper.

The metareviewer sincerely thanks the authors and reviewers for engaging into fruitful discussions.